# GR-Gaussian: Graph-Based Radiative Gaussian Splatting for Sparse-View Tomographic Reconstruction

## Abstract

Computed tomography (CT) reconstruction under sparse-view settings remains highly challenging due to severe artifacts. Recently, 3D Gaussian Splatting (3DGS) has shown promise for this task, but existing methods often rely on view-averaged gradient magnitudes, which easily cause needle-like artifacts in sparse views. To overcome this limitation, we propose GR-Gaussian, a graph-based 3DGS framework. It explicitly leverages a CT-specific prior, where regions of the same tissue or material have similar attenuation coefficients, forming a natural structural relationship among neighboring points. This structure motivates a graph-based representation, which guides gradient refinement to suppress needle-like artifacts. To exploit this structure, GR-Gaussian introduces (1) a Denoised Point Cloud Initialization strategy that mitigates initialization errors, and (2) a Pixel-Graph-Aware Gradient strategy that leverages graph-based density differences to refine gradient computation, improving splitting accuracy and density representation. Experiments on X-3D and real-world datasets validate the effectiveness of GR-Gaussian, achieving PSNR improvements of **0.67 dB** and **0.92 dB**, and SSIM gains of **0.011** and **0.021**. These results highlight the importance of embedding domain-specific structural priors for accurate CT reconstruction under challenging sparse-view conditions.

## 1 Introduction

Computed Tomography (CT) is a widely used non-invasive imaging technique applied in fields such as medicine, biology, security, and industry. It utilizes X-rays to capture multi-angle 2D projections, which measure the attenuation of rays as they pass through an object(Kak & Slaney, 2001). These projections are then processed to reconstruct the 3D density distribution of the object, enabling detailed internal visualization. In many applications, tomography data acquisition is limited by the number of projections or non-uniform angular sampling. For instance, medical CT reduces the number of projections to minimize X-ray exposure, while electron tomography faces the missing wedge problem due to restricted rotation angles. These constraints make the reconstruction problem highly ill-posed.Traditional methods like FBP and FDK(Feldkamp et al., 1984) are efficient but prone to artifacts, while model-based(Andersen & Kak, 1984) and deep learning approaches(Lin et al., 2023; Chung et al., 2023; Liu et al., 2023) improve accuracy but are computationally expensive and require large datasets.

The emergence of Neural Radiance Fields (NeRF)(Mildenhall et al., 2021) has provided a new paradigm for "per-case" tomographic reconstruction. NeRF implicitly models the 3D density field and radiance field through a neural network and optimizes the 3D structure from projection data combined with volume rendering technology. It does not rely on large-scale datasets and can retain fine structural details(Zha et al., 2022; Cai et al., 2024b; Zang et al., 2021). However, this method requires intensive point sampling for each ray during volume rendering, resulting in excessively long single reconstruction times. Even on high-performance computing devices, reconstructing a medium-resolution 3D model can take several hours. This severely limits its application in scenarios with high real-time requirements, such as intraoperative CT navigation.

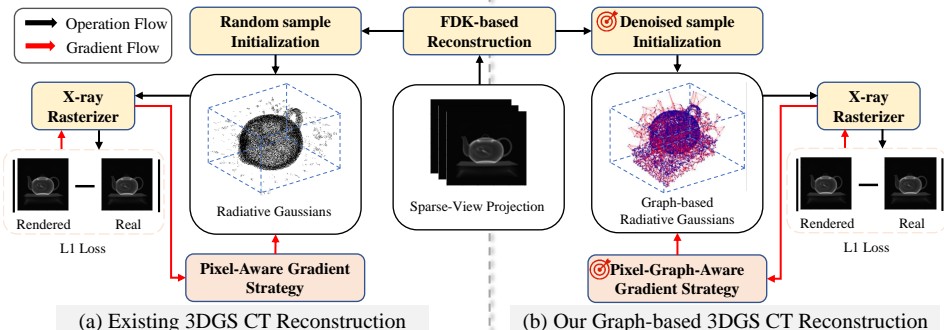

(a) Existing 3DGS CT Reconstruction     (b) Our Graph-based 3DGS CT Reconstruction

Figure 1: Comparison between existing 3DGS CT reconstruction method applying random-initialized and Pixel-Aware Gradient strategy for 3DGS in (a) and our proposed denoise sanmple initialization and Pixel-Graph-Aware Gradient strategy for 3DGS in (b).

Recently, 3D Gaussian Splatting (3DGS) (Kerbl et al., 2023; Yu et al., 2024; Guédon & Lepetit, 2024) has demonstrated superior quality and efficiency over NeRF in view synthesis tasks (Lu et al., 2024; Liang et al., 2024). Inspired by this, recent studies, such as X-Gaussian (Cai et al., 2024a) and $R^2$-Gaussian (Zha et al., 2024), have extended 3DGS to X-ray imaging for CT reconstruction. A central challenge in sparse-view CT reconstruction with 3DGS lies in both initialization and update. In existing approaches (Fig. 1 (a)), initialization is often performed by randomly sampling point coordinates and density values, which lacks structural guidance and leads to a mismatch with the underlying anatomy. Since initialization largely determines the stability and convergence of subsequent optimization, several alternatives—such as random, spherical, cubic, or FDK-guided sampling—have been explored (Fig. 2). Among these, FDK-based initialization provides the strongest starting point, but under sparse-view conditions, FDK reconstructions contain severe artifacts that distort initial coordinates and densities, ultimately degrading reconstruction quality. This limitation underscores the need for a more robust initialization strategy that can preserve structural consistency even when the number of views is severely limited.

Following the initialization challenges, the update process in existing 3DGS-based methods is further constrained by pixel-aware gradients, which rely solely on point-to-pixel correspondences. For flat Gaussian ellipsoids, the projected area varies with viewing angle, and sparse-view sampling captures only a few extreme cases. This leads to biased gradient estimates that fail to effectively update irregular regions, such as bone edges or tissue outliers. Sparse data further amplify these issues: large Gaussian kernels are inadequately split, disrupting inter-point relationships and producing needle-like artifacts. Together, these limitations leave critical structures under-optimized, compromising overall reconstruction quality and motivating the need for a more robust gradient update strategy under sparse-view conditions.

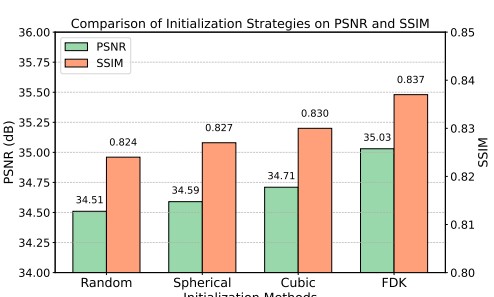

Figure 2: Comparison of the different initialization strategies for CT reconstruction in Real-world dataset. We see the FDK enhances results.

Motivated by the limitations of existing sparse-view CT reconstruction methods, we introduce GR-Gaussian, a graph-based 3DGS framework (Fig. 1 (b)) with two key innovations. First, we introduce a Denoised Point Cloud Initialization strategy (De-Init) that enhances traditional FDK-based initialization by removing noise and artifacts, producing more accurate initial points. This is critical under sparse-view conditions, where FDK reconstructions often contain severe artifacts that misalign coordinates and degrade subsequent updates, especially in real-world noisy environments. Second, we propose a Pixel-Graph-Aware Gradient strategy (PGA) that leverages inter-point relationships to guide updates, regulating the spatial distribution of points and mitigating needle-like artifacts. To-

gether, these strategies improve reconstruction quality under sparse views. Additionally, we incorporate an efficient optimization scheme with adaptive density control to ensure stable performance across challenging scenarios.

We validate GR-Gaussian through extensive experiments on both the simulated X-3D dataset(Zha et al., 2024) and real-world CT datasets(Society, 2024), comparing it against multiple baselines. The results show that our framework consistently improves reconstruction quality, achieving PSNR gains of **0.67 dB** and **0.92 dB**, and SSIM improvements of **0.011** and **0.021**. Our main contributions are summarized as follows: ♠ We propose a novel framework GR-Gaussian, which leverages graph-structured relationships to model objects, providing a solution for sparse-view tomographic reconstruction. ♡ To address challenges in sparse-view 3DGS reconstruction, we introduce De-Init to improve initialization and PGA to refine gradient computation, effectively reducing needle-like artifacts and improving reconstruction quality. ♣ Our framework achieves state-of-the-art performance on the X-3D and real-world datasets, demonstrating its effectiveness under challenging sparse-view conditions.

## 2 RELATED WORK

**Computed Tomography (CT)** is widely used in medicine(Hounsfield, 1980; Katsuragawa & Doi, 2007) and industry(De Chiffre et al., 2014; Zang et al., 2019). Sparse-view CT reconstruction is challenging due to limited projections. Analytical methods, such as Filtered Back Projection (FBP) and its 3D counterpart FDK(Feldkamp et al., 1984), are efficient but prone to severe streak artifacts. Iterative methods(Andersen & Kak, 1984; Sidky & Pan, 2008; Manglos et al., 1995; Sauer & Bouman, 2002) reduce artifacts via regularized energy minimization but are computationally expensive and may lose fine structural details. Deep learning approaches(Lin et al., 2023; Chung et al., 2023; Liu et al., 2023) show promise: supervised methods learn semantic priors for projection inpainting(Anirudh et al., 2018; Ghani & Karl, 2018) or volume denoising(Chung et al., 2023; Lee et al., 2023; Liu et al., 2023; 2020) but struggle to generalize, while self-supervised methods inspired by NeRF(Mildenhall et al., 2021) optimize density fields using photometric losses(Cai et al., 2024b; Zha et al., 2022; Zang et al., 2021) at high computational cost.

**3D Gaussian Splatting (3DGS)** has achieved remarkable results in RGB tasks, including surface reconstruction(Li et al., 2025; Lin et al., 2025; Hou et al., 2025), dynamic scene modeling(Zhu et al., 2025; Huang et al., 2025; Wu et al., 2024), human avatar creation(Qu et al., 2025; Li et al., 2024), and 3D generation, outperforming NeRF in speed via parallelized rasterization. Inspired by these successes, recent studies have explored 3DGS for X-ray imaging: X-Gaussian(Cai et al., 2024a) synthesizes novel-view X-ray projections mainly for data augmentation, while $R^2$-Gaussian(Zha et al., 2024) represents density fields with customized Gaussian kernels but replaces efficient rasterization with CT simulators. Although these approaches can fit projection data, they primarily consider the task as a "surface fitting" problem transferred from RGB applications and do not fully exploit the internal structural information inherent in CT volumes. As a result, they often overfit sparse-view projections, motivating methods that explicitly model internal density relationships to robustly handle sparse-view CT reconstruction.

## 3 PRELIMINARY

**Radiative Gaussians.** The previous approach (Zha et al., 2024) employs learnable 3D Gaussian kernels, $\mathbb{G}^3 = \{G_i^3\}_{i=1,\cdots,M}$, termed Radiative Gaussians, to represent the target object. Each $G_i^3$ models a local density field as $G_i^3(\mathbf{x} \mid \rho_i, \mathbf{p}_i, \Sigma_i) = \rho_i \cdot e^{-\frac{1}{2}(\mathbf{x}-\mathbf{p}_i)^\top \Sigma_i^{-1}(\mathbf{x}-\mathbf{p}_i)}$, where $\rho_i$ is the central density, $\mathbf{p}_i \in \mathbb{R}^3$ is the position, and $\Sigma_i \in \mathbb{R}^{3\times3}$ is the covariance matrix (decomposed as $\Sigma_i = \mathbf{R}_i \mathbf{S}_i \mathbf{S}_i^\top \mathbf{R}_i^\top$ with rotation $\mathbf{R}_i$ and scale $\mathbf{S}_i$). These Gaussians may exhibit flattened long-axis shapes depending on $\Sigma_i$'s scale factors. The total density is $\sigma(\mathbf{x}) = \sum_{i=1}^{M} G_i^3(\mathbf{x} \mid \rho_i, \mathbf{p}_i, \Sigma_i)$. In computed tomography, the forward projection follows the Beer–Lambert law, $I = I_0 \exp(-\int \sigma(\mathbf{x})\,ds)$, where $I_0$ denotes the incident intensity and $ds$ the differential ray path. Unlike RGB rendering, CT relies on isotropic X-ray attenuation without view-dependent color, making Radiative Gaussians a natural fit for modeling volumetric density. The overall process is illustrated in Fig. 3 (a).

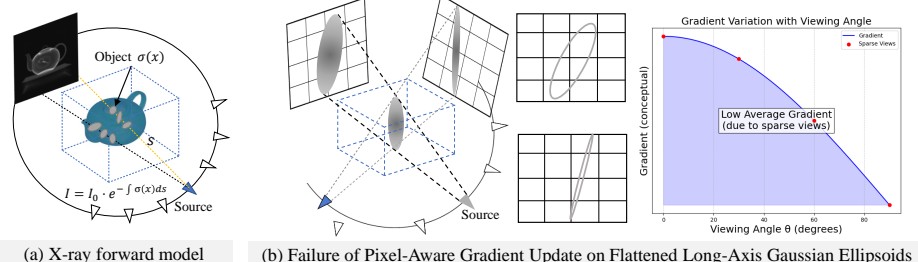

(a) X-ray forward model      (b) Failure of Pixel-Aware Gradient Update on Flattened Long-Axis Gaussian Ellipsoids

Figure 3: (a) X-ray forward model. (b) Failure of Pixel-Aware Gradient Update on Flattened Long-Axis Gaussian Ellipsoids.

**Pixel-Aware Gradient.** To optimize Radiative Gaussians in CT reconstruction, we adopt a pixel-aware gradient strategy inspired by 3D Gaussian Splatting. In this setting, the decision to split or clone a Gaussian kernel $G_i^3$ relies on the average gradient magnitude of its projected Normalized Device Coordinates (NDC) across viewpoints $v$. A kernel is refined when

$$\frac{1}{N^i} \sum_{v=1}^{N^i} \left\| \frac{\partial L_v}{\partial \mu_{\text{ndc}}^{i,v}} \right\|_2 > \tau_{\text{pos}}, \tag{1}$$

with Adaptive Density Control (ADC) further adjusting kernel density every 100 iterations. In CT reconstruction, the grayscale nature of images simplifies gradient computation by removing channel-wise summation. The contribution of a pixel under viewpoint $v$ to Gaussian $G_i^3$ is given by

$$(\mathbf{g}_i^{c'})^v = \sum_{pix=1}^{m_v^i} \frac{\partial L_v}{\partial \alpha_{v,pix}^i} \cdot \frac{\partial \alpha_{v,pix}^i}{\partial \mu_{\text{ndc}}^{i,v}}, \tag{2}$$

where $\mu_{\text{ndc}}^{i,v}$ are the projected NDC coordinates and $\alpha_{v,pix}^i$ denotes the pixel's density contribution. However, under sparse-view conditions, flattened long-axis Gaussians often yield low average gradients due to limited angular coverage, as illustrated in Fig. 3 (b). This failure mode prevents accurate updates in anomalous regions, motivating our proposed gradient refinement strategy.

## 4 METHOD

In this section, we present GR-Gaussian, a novel framework that represents objects through graph-structured relationships (Sec. 4.1). Specifically optimized for sparse-view CT reconstruction, our framework effectively tackles critical issues, most notably the emergence of needle-like artifacts (Sec. 4.2). We first introduce a Denoised Point Cloud Initialization Strategy based on an enhanced FDK method (Sec. 4.2.1). Subsequently, a Pixel-Graph-Aware Gradient Strategy is proposed to refine the gradient computation process (Sec. 4.2.2). Finally, we detail an optimization strategy specifically tailored for GR-Gaussian (Sec. 4.2.3).

### 4.1 REPRESENTING OBJECTS WITH GRAPH-BASED RADIATIVE GAUSSIANS

As shown in Fig. 4, an object is represented as a learnable 3D Gaussian graph $\mathcal{G}$, defined as:

$$\mathcal{G} = (\mathcal{V}, \mathcal{E}, \{G_i^3(\rho_i, \mathbf{p}_i, \Sigma_i)\}_{i=1}^M), \tag{3}$$

where $\mathcal{V} = \{\mathbf{p}_i\}_{i=1}^M$ denotes the positions of the Gaussian kernels, $\mathcal{E}$ represents the edges connecting the kernels, and $G_i^3$ is a Gaussian kernel parameterized by density $\rho_i$, position $\mathbf{p}_i \in \mathbb{R}^3$, and covariance $\Sigma_i \in \mathbb{R}^{3 \times 3}$.

The graph structure $\mathcal{G}$ is constructed to capture spatial relationships between Gaussian kernels. The vertices $\mathcal{V}$ represent kernel positions, while the edges $\mathcal{E}$ are determined using the KNN algorithm:

$$\text{KNN}(\mathbf{p}_i) = \{\mathbf{p}_j \mid j \in \arg\min_j d(\mathbf{p}_i, \mathbf{p}_j), j = 1, \cdots, K\}, \tag{4}$$

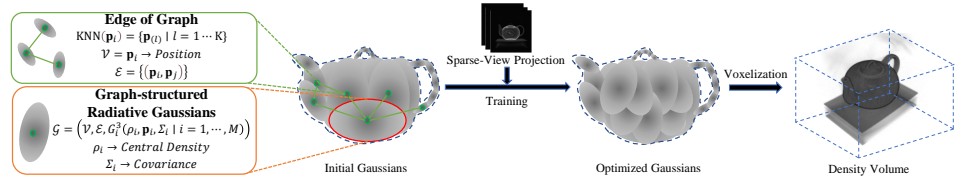

Figure 4: The scanned object is represented as **graph-based radiative Gaussians**, optimized using real X-ray projections to retrieve the density volume via voxelization.

where the Euclidean distance $d(\mathbf{p}_i, \mathbf{p}_j)$ is given by:

$$d(\mathbf{p}_i, \mathbf{p}_j) = \|\mathbf{p}_i - \mathbf{p}_j\|_2. \tag{5}$$

To ensure bidirectional connectivity, the edge set $\mathcal{E}$ is defined as:

$$\mathcal{E} = \{(\mathbf{p}_i, \mathbf{p}_j) \mid \mathbf{p}_j \in \text{KNN}(\mathbf{p}_i) \text{ and } \mathbf{p}_i \in \text{KNN}(\mathbf{p}_j)\}. \tag{6}$$

Each Gaussian kernel $G_i^3$ defines a local density field by combining its own density with contributions from neighboring kernels:

$$G_i^3(\mathbf{x}) = \rho_i \cdot G(\mathbf{x} \mid \mathbf{p}_i, \Sigma_i), \tag{7}$$

The combined density $D(\mathbf{x})$ at a position $\mathbf{x}$ is computed as:

$$D(\mathbf{x}) = \sum_{i=1}^{M} G_i^3(\mathbf{x}). \tag{8}$$

This graph-based representation is specifically designed to model the relationships between neighboring points, enabling the incorporation of inter-point information. By leveraging these relationships, the framework facilitates the optimization of update strategies, improving the accuracy of the reconstruction process.

## 4.2 TRAINING GRAPH-BASED RADIATIVE GAUSSIANS

Our training pipeline, illustrated in Fig. 5 (a), begins with the initialization of Graph-based Radiative Gaussians from a modified FDK volume. A graph is then constructed using the KNN method to capture spatial relationships between Gaussian kernels. Projections are rasterized to compute photometric losses, while tiny density volumes are voxelized for 3D regularization. Finally, a modified Pixel-Graph-Aware Gradient Strategy is applied to densify the Gaussians.

### 4.2.1 DENOISED POINT CLOUD INITIALIZATION STRATEGY

Previous methods utilize FDK to generate low-quality volumes for initialization. However, under sparse-view conditions, FDK reconstructions suffer from significant noise and artifacts, which degrade subsequent processing. To address this, as shown in Fig. 5 (b), we propose a Denoised Point Cloud Initialization Strategy leveraging 3DGS characteristics. Gaussian filtering is applied to the FDK-reconstructed point cloud to suppress noise and artifacts while preserving structural details, ensuring high-quality initialization for robust and accurate reconstruction. The Gaussian filtering process is defined as:

$$f'(i) = \frac{1}{Z} \sum_{j \in \mathcal{N}(i)} G(j; \sigma_d) \cdot f(i+j), . \tag{9}$$

Here, $i$ denotes the coordinates of the current voxel, and $j$ represents the offset within its neighborhood $\mathcal{N}(i)$, defined as $j_k \in [-r, r]$ for all dimensions $k$. The Gaussian kernel is given by:

$$G(j; \sigma_d) = \exp\left(-\frac{|j|^2}{2\sigma_d^2}\right), \tag{10}$$

where $|j|^2 = j_1^2 + j_2^2 + \cdots + j_n^2$, and $Z = \sum_{j \in \mathcal{N}(i)} G(j; \sigma_d)$ is the normalization factor ensuring the kernel weights sum to 1. Empty regions are excluded using a density threshold $\tau$, and $M$ points are randomly sampled as kernel positions. Gaussian scales are set based on nearest neighbor distances, assuming no rotation. Central densities are directly queried from the denoised volume.

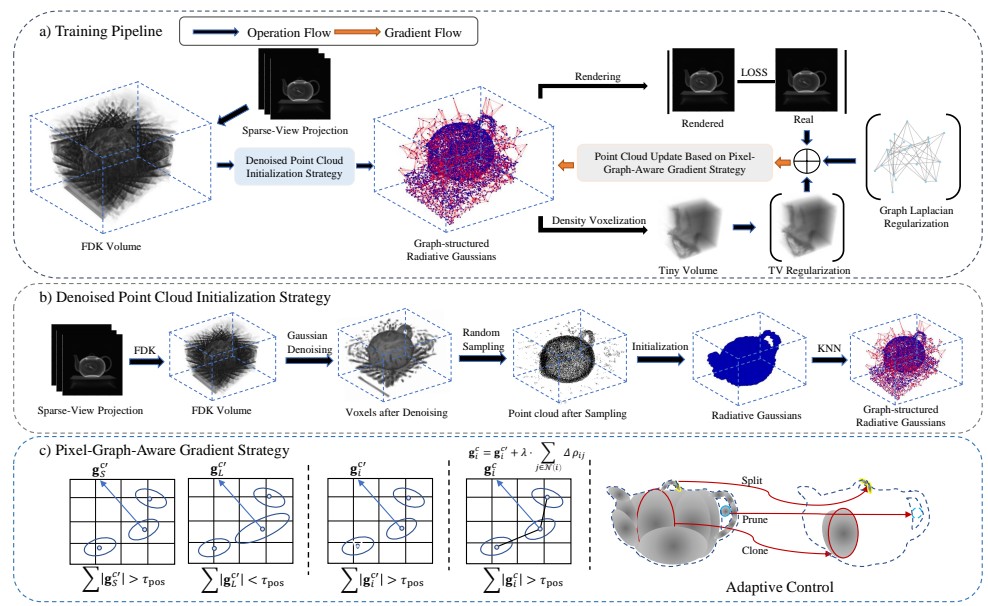

Figure 5: **Training pipeline of GR-Gaussian**. (a) Overall training pipeline. (b) Denoised Point Cloud Initialization Strategy. (c) Pixel-Graph-Aware Gradient Strategy.

### 4.2.2 PIXEL-GRAPH-AWARE GRADIENT STRATEGY

Previous methods compute gradients based solely on average values, which limits the contribution of large kernels with small gradients to the densification process. To address this limitation, as shown in Fig. 5 (c), we propose a **Pixel-Graph-Aware Gradient Strategy** that adjusts the magnitude of gradients used in densification by incorporating graph-based density variations.

Specifically, for each Gaussian kernel, we measure the density differences with its graph-connected neighbors, consistent with the characteristics of X-ray CT imaging, where tissues or materials within the same region typically exhibit nearly constant attenuation coefficients. The density difference between Gaussian $i$ and its neighbor $j$ is defined as

$$\Delta \rho_{ij} = |\rho_i - \rho_j|, \quad \forall j \in \mathcal{N}(i). \tag{11}$$

These density variations serve as an additional weighting factor to enlarge the effective gradient magnitude for Gaussians located at structural boundaries. For a Gaussian kernel $G_i^3$ under viewpoint $v$, the augmented gradient magnitude is computed as

$$\|(\mathbf{g}_i^c)^v\|_{\mathrm{aug}} = \|(\mathbf{g}_i^{c'})^v\| + \lambda_g \cdot \sum_{j \in \mathcal{N}(i)} \Delta \rho_{ij}, \tag{12}$$

where $\lambda_g$ controls the contribution of neighboring kernels and $\mathcal{N}(i)$ is the neighbor set of $G_i^3$. By weighting gradients with local density differences, PGA ensures that Gaussians with significant structural variation are more likely to exceed the splitting condition

$$\|(\mathbf{g}_i^c)^v\|_{\mathrm{aug}} > \tau_{\mathrm{pos}}, \tag{13}$$

where $\tau_{\mathrm{pos}}$ is a predefined threshold. This improves densification around structural boundaries while leaving the standard parameter update process unchanged.

### 4.2.3 DENSITY VOXELIZER AND OPTIMIZATION

We employ a voxelizer $\mathbf{V}$ to convert the Gaussian representation $\mathcal{G}$ into a voxel volume $V \in \mathbb{R}^{X \times Y \times Z}$, $V = \mathbf{V}(\mathcal{G})$. The voxelizer partitions the space into $8 \times 8 \times 8$ tiles and performs Gaussian culling, retaining kernels with 99% confidence of intersecting each tile. To incorporate graph information, we weight each Gaussian's contribution to voxels by the density differences with its neighboring kernels in the graph. This enhances the influence of Gaussians near structural boundaries

during accumulation, without modifying the voxel selection or accumulation logic. The resulting voxel density is then used for rendering and gradient computation in 3D Gaussian Splatting.

To optimize the radiative Gaussians, we employ Adam. The total loss function consists of the photometric loss $\mathcal{L}_1$, the D-SSIM loss $\mathcal{L}_{ssim}$(Wang et al., 2004), the 3D total variation (TV) regularization $\mathcal{L}_{tv}$, and the graph Laplacian regularization $\mathcal{L}_{lap}$. The graph Laplacian regularization term $\mathcal{L}_{lap}$ is defined as:

$$\mathcal{L}_{lap}(\mathcal{G}) = \sum_{i=1}^{M} \sum_{j \in \mathcal{N}(i)} w_{ij}(\rho_i - \rho_j)^2, \tag{14}$$

where $\mathcal{G}$ represents the graph structure of the Gaussian kernels, $\rho_i$ and $\rho_j$ denote the densities of Gaussian kernels $G_i^3$ and $G_j^3$, and $w_{ij}$ is the weight of the edge $(i, j)$. This term encourages local smoothness by minimizing density differences between neighboring kernels while preserving boundary information. To further enhance regularization, we define $\mathcal{L}_{norm}$ as:

$$\mathcal{L}_{norm} = \lambda_{lap}\mathcal{L}_{lap}(\mathcal{G}) + \lambda_{tv}\mathcal{L}_{tv}(\mathcal{V}_{tv}), \tag{15}$$

where $\mathcal{L}_{tv}$ represents the 3D total variation regularization applied to the volume $\mathcal{V}_{tv}$. The total loss function is then expressed as:

$$\mathcal{L}_{total} = \mathcal{L}_1(\mathbf{I}_r, \mathbf{I}_m) + \lambda_{ssim}\mathcal{L}_{ssim}(\mathbf{I}_r, \mathbf{I}_m) + \mathcal{L}_{norm}. \tag{16}$$

Here, $\mathbf{I}_r$ and $\mathbf{I}_m$ denote the reconstructed and measured images, respectively.

Table 1: Quantitative results. We colorize the best , second-best , and third-best numbers.

| Method | HO | | BS | | AO | | Average | | RD | |
|---|---|---|---|---|---|---|---|---|---|---|
| | PSNR | SSIM | PSNR | SSIM | PSNR | SSIM | PSNR | SSIM | PSNR | SSIM |
| FDK | 22.64 | 0.319 | 23.45 | 0.296 | 22.95 | 0.336 | 23.01 | 0.317 | 23.30 | 0.335 |
| SART | 29.58 | 0.773 | 32.37 | 0.878 | 31.48 | 0.825 | 31.14 | 0.825 | 31.52 | 0.790 |
| ASD-POCS | 29.42 | 0.810 | 31.40 | 0.887 | 30.58 | 0.845 | 30.47 | 0.847 | 31.32 | 0.810 |
| NAF | 32.05 | 0.841 | 34.36 | 0.930 | 35.34 | 0.909 | 33.92 | 0.893 | 32.92 | 0.772 |
| SAX-NeRF | 32.53 | 0.858 | 34.67 | 0.940 | 35.85 | 0.917 | 34.35 | 0.905 | 33.49 | 0.793 |
| $R^2$-GS | 32.98 | 0.881 | 35.08 | 0.944 | 37.52 | 0.945 | 35.19 | 0.922 | 35.03 | 0.837 |
| Ours | **33.47** | **0.891** | **35.56** | **0.952** | **38.55** | **0.955** | **35.86** | **0.933** | **35.95** | **0.858** |

## 5 EXPERIMENTS

### 5.1 EXPERIMENTAL SETTINGS

**Dataset**. Following $R^2$-Gaussian , we utilize the publicly available X-3D dataset to evaluate our method. This dataset(Zha et al., 2024) encompasses diverse categories, including human organs(chest, foot, head, jaw, and pancreas), artificial objects(backpack, engine, mount, present, and teapot), and biological specimens(beetle, bonsai, broccoli, kingsnake, and pepper), providing a comprehensive benchmark for reconstruction performance. Using the tomography toolbox TIGRE, we generate $512 \times 512$ projections along a full circular trajectory, incorporating electric noise and ponton scatter to simulate realistic conditions. Subsequently, CT volumes are scanned within an angular range of $0°$ to $360°$ to produce 25-view projections. For the Real-World datasets(Society, 2024), the projections are directly acquired using CT scanners and correspond to specific angles. Each projection has a resolution of $560 \times 560$. To achieve sparse-view conditions, we uniformly select 25 projection angles from the full angular range of $0°$ to $360°$. This dataset including pine, seashell and walnut.

**Implementation Details**. The GR-Gaussian framework is implemented using PyTorch(Paszke, 2019), PyTorch Geometric(Fey & Lenssen, 2019), and CUDA(Guide, 2013), and optimized with the Adam optimizer(Diederik, 2014). Initial learning rates for location, density, scale, and rotation are set to 0.0002, 0.01, 0.005, and 0.001, respectively, with exponential decay reducing them to 10% of their initial values. Regularization parameters include a TV volume level of $D = 32$, loss weights $\lambda_{ssim} = 0.25$ and $\lambda_{tv} = 0.05$, a Graph Laplacian weight $\mathcal{L}_{lap} = 8 \times 10^{-4}$, a Gaussian denoising parameter $\sigma_d = 3$, and a gradient computation weight $\lambda_g = 1 \times 10^{-4}$. The framework uses

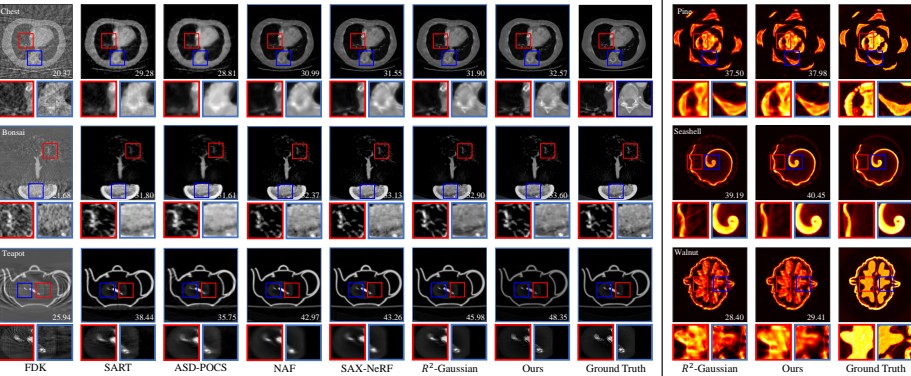

Figure 6: The left shows **CT reconstructions from the X-3D dataset**, covering three categories: chest, bonsai, and teapot. The right displays **Real-world dataset reconstructions** with colorized slices to highlight details, all under 25-view conditions.

$M = 50,000$ Gaussians, a density threshold $\tau = 0.001$, and $k = 6$ nearest neighbors. A dynamic stopping criterion ($Iter_{stop}$) evaluates PSNR every 500 iterations, terminating if PSNR decreases by more than 0.5%. All experiments were conducted on an RTX 4090 GPU. Quantitative evaluation employed PSNR and SSIM(Wang et al., 2004), with PSNR assessing 3D reconstruction accuracy and SSIM evaluating structural consistency across 2D slices.

## 5.2 RESULTS AND EVALUATION

The experiments assess GR-Gaussian for sparse-view tomographic reconstruction on both X-3D and real-world datasets. On X-3D, it surpasses the SOTA 3DGS baseline (Zha et al., 2024), NeRF-based approaches (Zha et al., 2022; Cai et al., 2024b), and classical algorithms (Feldkamp et al., 1984; Andersen & Kak, 1984; Sidky & Pan, 2008), with an average gain of **0.67 dB** in PSNR and **0.011** in SSIM (Table 1). On real-world CT scans, GR-Gaussian achieves **0.92 dB** higher PSNR and **0.021** higher SSIM, showing resilience to noise and acquisition artifacts. Figure 6 further illustrates that the method produces sharper edges and fewer streak patterns than competing baselines. Notably, in the *teapot* case, PSNR improves from **45.98 dB** to **48.35 dB**, while in the *seashell* case with thin ridges, PSNR increases from **39.19 dB** to **40.45 dB**. These results indicate that GR-Gaussian consistently improves reconstruction fidelity, with clear gains on real CT data where sparse-view sampling and measurement noise limit existing methods.

## 5.3 ABLATION STUDY

**Component Analysis**. We evaluate the contributions of De-Init, PGA, and graph Laplacian regularization (Reg) on the X-3D and real-world datasets under 25-view sparse conditions. The baseline uses FDK-based initialization and standard gradient updates, omitting all three components. Table 2 reports PSNR, SSIM, and runtime; Fig. 7(a) shows qualitative comparisons. **De-Init** reduces initialization errors, suppressing needle-like artifacts and providing a cleaner starting point. **PGA** refines gradients, improving smooth regions and structural consistency by correcting anomalous points during updates. Their effects partially overlap: De-Init prevents errors, PGA updates them. **Reg** alone provides limited gain, showing that smoothness without proper initialization or gradient-aware updates is insufficient.

Table 2: Ablation study on X-3D and Real-world datasets.

| Config. | X-3D | | | Real-world | | |
|---|---|---|---|---|---|---|
| | PSNR | SSIM | T(s) | PSNR | SSIM | T(s) |
| Baseline (B) | 35.20 | 0.923 | 508 | 34.93 | 0.835 | 576 |
| B + Reg | 35.30 | 0.925 | 523 | 35.28 | 0.842 | 584 |
| B + De-Init | 35.61 | 0.931 | 583 | 35.90 | 0.857 | 643 |
| B + PGA | 35.77 | 0.932 | 603 | 35.62 | 0.848 | 639 |
| Full model | 35.86 | 0.932 | 629 | 35.95 | 0.858 | 669 |

Table 3: Performance comparison for k.

|  | k | 4 | 5 | 6 | 7 | 8 |
|---|---|---|---|---|---|---|
|  | **PSNR** | 35.80 | 35.84 | 35.86 | 35.87 | 35.89 |
| X-3D | **SSIM** | 0.932 | 0.932 | 0.933 | 0.933 | 0.934 |
|  | **Time** | 536s | 567s | 629s | 763s | 865s |
|  | **PSNR** | 35.82 | 35.87 | 35.95 | 35.97 | 36.01 |
| Real World | **SSIM** | 0.851 | 0.854 | 0.858 | 0.858 | 0.860 |
|  | **Time** | 562s | 597s | 669s | 794s | 927s |

Table 4: Performance comparison for $\sigma_d$.

|  | $\sigma_d$ | 1 | 2 | 3 | 4 | 5 |
|---|---|---|---|---|---|---|
|  | **PSNR** | 35.60 | 35.72 | 35.86 | 35.68 | 35.43 |
| X-3D | **SSIM** | 0.926 | 0.928 | 0.933 | 0.927 | 0.925 |
|  | **Time** | 589s | 613s | 629s | 633s | 645s |
|  | **PSNR** | 35.90 | 35.93 | 35.95 | 35.91 | 35.85 |
| Real World | **SSIM** | 0.857 | 0.858 | 0.858 | 0.857 | 0.856 |
|  | **Time** | 641s | 655s | 669s | 675s | 682s |

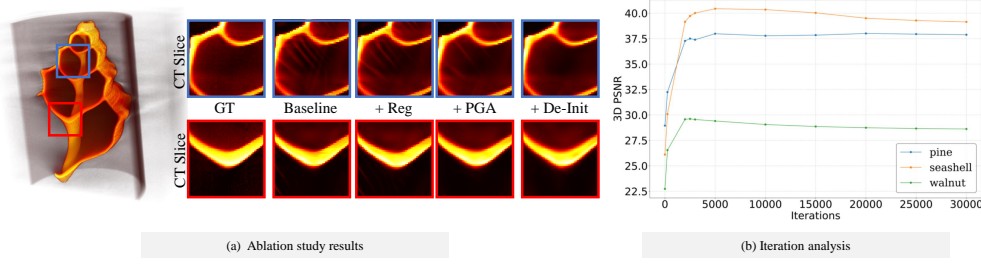

(a) Ablation study results

(b) Iteration analysis

Figure 7: (a) Ablation study. The results highlight the contributions of PGA and De-Init in improving reconstruction quality. (b) Iteration analysis. Reconstruction results of GR-Gaussian at different iteration counts, illustrating how iteration number affects PSNR.

The full GR-Gaussian framework, combining De-Init and PGA, achieves the highest PSNR and SSIM across both datasets. Qualitative results in Fig. 7 (a) show that De-Init mitigates sharp artifacts, while PGA preserves smooth structures and fine details. These observations align with quantitative metrics, demonstrating that the combined strategy effectively balances artifact suppression and structural fidelity.

**Parameter Analysis**. We evaluate the effects of key hyperparameters and iteration behavior under 25-view sparse conditions. Table 3 and 4 show that reconstruction quality peaks at $k = 6$ and $\sigma_d = 3$. Larger $k$ slightly improves PSNR/SSIM by capturing more neighbors but increases runtime, while smaller $k$ reduces computation at the cost of gradient precision. For $\sigma_d$, smaller values preserve fine details but leave noise, whereas larger values reduce noise but can over-smooth edges.

**Iteration Analysis**. Under sparse-view conditions, PSNR initially improves with increased iterations but eventually declines (see Fig. 7 (b)). This reduction is less pronounced in the simulated X-3D dataset, yet it becomes evident in real-world datasets, where noise and artifacts exacerbate reconstruction degradation. To address this issue, we employ a dynamic iteration stopping criterion ($Iter_{stop}$), where PSNR is evaluated every 500 iterations. If the PSNR decreases by more than 0.5%, the iteration process is terminated. This approach ensures that the model avoids overfitting and maintains optimal reconstruction quality.

# 6 CONCLUSION

This paper introduces GR-Gaussian, a novel framework based on 3D Gaussian Splatting for sparse-view tomographic reconstruction. Our approach employs a Denoised Point Cloud Initialization Strategy that robustly reduces noise while preserving critical structural details. Furthermore, by incorporating a graph-structured representation together with a novel gradient computation mechanism, GR-Gaussian significantly enhances reconstruction quality and effectively mitigates artifacts under challenging sparse-view conditions. Extensive evaluations on X-3D and real-world datasets demonstrate that our method outperforms existing techniques in both artifact suppression and reconstruction accuracy.

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

Table 5: GR-Gaussian Framework Parameter Settings

| Parameter | Value |
|---|---|
| Location learning rate | 0.0002 |
| Density learning rate | 0.01 |
| Scale learning rate | 0.005 |
| Rotation learning rate | 0.001 |
| TV Volume Level ($D$) | 32 |
| SSIM Loss Weight ($\lambda_{\text{ssim}}$) | 0.25 |
| TV Loss Weight ($\lambda_{\text{tv}}$) | 0.05 |
| Graph Laplacian Regularization ($\mathcal{L}_{\text{lap}}$) | $8 \times 10^{-4}$ |
| Gradient Computation Weight ($\lambda_{\text{g}}$) | $1 \times 10^{-4}$ |
| Number of Gaussians ($M$) | 50,000 |
| Density Threshold ($\tau$) | 0.001 |
| Gaussian Denoising Parameter ($\sigma_d$) | 3 |
| Number of nearest neighbors in graph ($k$) | 6 |

## A  ETHICS STATEMENT AND REPRODUCIBILITY STATEMENT

**Ethics Statement.** This work focuses on developing methods for sparse-view tomographic reconstruction, with potential applications in medical imaging and industrial inspection. While improved reconstruction quality can benefit clinical diagnosis and scientific analysis, we emphasize that the method is a technical contribution and should not be interpreted as a direct clinical solution without further validation. No human or animal data were collected for this study; only publicly available and synthetic datasets were used. We are committed to ensuring that the proposed method is applied responsibly, respecting privacy, safety, and ethical standards in its downstream applications.

**Reproducibility Statement.** We make every effort to ensure reproducibility of our results. All implementation details, including hyperparameter settings, optimization schedules, and evaluation metrics, are described in the main paper and appendix. The datasets used in this work are publicly available, and we will release the source code, pretrained models, and scripts for data preprocessing, training, and evaluation upon publication. This allows independent researchers to reproduce our results and build upon our work.

## B  APPENDIX: AI TOOLS

We used a large language model (GPT-5-mini) to assist in editing the Introduction and Related Work sections for clarity and readability. No technical content, experimental results, or scientific claims were generated or altered by the LLM. The LLM was used strictly for linguistic refinement; all scientific content was verified by the authors.

## C  IMPLEMENTATION DETAILS

Our GR-Gaussian framework is implemented in PyTorchPaszke (2019), a deep learning library widely used for flexible and efficient model development. Graph-based operations are implemented using PyTorch GeometricFey & Lenssen (2019), and CUDAGuide (2013) is used for parallel computation on NVIDIA GPUs. The model is trained using the Adam optimizerDiederik (2014).

### C.1  PARAMETER SETTINGS

Following Zha et al. (2024), we adopt the following parameters for all experiments unless otherwise noted. These settings include learning rates for different Gaussian attributes, regularization weights, and graph-related parameters, as summarized in Table 5.

---

**Algorithm 1** GR-Gaussian Initialization Algorithm

---

**Input**: Sparse-view X-ray projections
**Parameters**: $\sigma_d, M, \tau, k$
**Output**: Initial 3DGS point cloud including location, density, scale, rotation, and graph $\mathcal{G}$

1: **FDK Reconstruction:** Generate coarse voxel data.
2: **Denoising:** Apply Gaussian filter with $\sigma_d$.
3: **Adaptive Sampling:**
4: **for** each voxel with density $\geq \tau$ **do**
5:    Add voxel location and density to candidate set
6: **end for**
7: Randomly select $M$ points from the candidate set to form initial point cloud.
8: **Graph Construction:** Build graph $\mathcal{G}$ using KNN with $k$ nearest neighbors.
9: **return** 3DGS point cloud with graph $\mathcal{G}$.

---

## C.2 Denoised Point Cloud Initialization Strategy

The Denoised Point Cloud Initialization Strategy (De-Init) addresses limitations of traditional FDK-based initialization under sparse-view conditions. Noise and artifacts in the initial volume can degrade reconstruction quality. De-Init integrates enhanced FDK reconstruction, Gaussian filtering, and adaptive sampling to generate a high-quality initial point cloud, providing a robust foundation for graph-structured radiative Gaussians in GR-Gaussian.

The process consists of four stages: (1) **FDK Reconstruction**, producing a coarse 3D voxel volume from sparse-view X-ray projections; (2) **Gaussian Denoising**, applying a 3D Gaussian filter with parameter $\sigma_d$ to suppress high-frequency noise while preserving structural details; (3) **Adaptive Sampling**, selecting voxels exceeding density threshold $\tau$ and randomly sampling $M$ points to form the initial Gaussian kernels; and (4) **Graph Construction**, building a graph $\mathcal{G}$ via KNN with $k$ nearest neighbors to model spatial correlations between kernels.

By combining denoising, adaptive sampling, and graph construction, De-Init reduces initialization errors, accelerates convergence, and ensures structural consistency aligned with graph Laplacian regularization in GR-Gaussian.

## C.3 Pixel-Graph-Aware Gradient Strategy

After initialization, the GR-Gaussian framework employs a structured training process to optimize the parameters of radiative Gaussian kernels, ensuring accurate and high-quality 3D tomographic reconstruction under sparse-view conditions. The framework utilizes a comprehensive loss function designed to balance pixel-wise accuracy, structural consistency, and smoothness in the reconstructed density field. Specifically, the $\mathcal{L}_1$ loss minimizes absolute differences between predicted and ground truth values, while the Structural Similarity Index Measure (SSIM) loss evaluates perceptual similarity to enhance structural coherence across 2D slices. To further refine the reconstruction, 3D Total Variation Regularization suppresses noise and enforces smooth transitions, and Graph Laplacian Regularization promotes local density smoothness while preserving critical boundary information to maintain structural integrity.

Gradients are computed using the Pixel-Graph-Aware Gradient Strategy, which incorporates density differences between neighboring kernels to refine gradient computation and improve optimization accuracy. This strategy also facilitates kernel splitting when gradient magnitudes exceed predefined thresholds, ensuring precise adjustments to the radiative Gaussian representation. To prevent overfitting and maintain reconstruction quality, a dynamic stopping criterion evaluates PSNR every 500 iterations, terminating the training process if PSNR decreases by more than 0.5%. All experiments were conducted on an NVIDIA RTX 4090 GPU, leveraging its computational efficiency to accelerate training. Quantitative evaluations employed PSNR to assess 3D reconstruction accuracy and SSIM to measure structural consistency across 2D slices, demonstrating the robustness and precision of the GR-Gaussian framework.

---

**Algorithm 2** GR-Gaussian Training Process

---

**Input**: Initialized 3DGS point cloud (location, density, scale, rotation, graph $\mathcal{G}$).
**Parameter**: Gradient threshold, loss weights, learning rates (see Table 5), update interval (100 iterations)
**Output**: Optimized 3DGS point cloud and density volumes

1: **Loss Function.** Define the total loss as a combination of:
- $\mathcal{L}_1$ Loss
- Structural Similarity Loss (SSIM)
- 3D Total Variation Regularization
- Graph Laplacian Regularization
2: **for** each iteration in training **do**
3:     **Gradient Optimization.** Compute gradients using Pixel-Graph-Aware Gradient Strategy, which incorporates density differences between neighboring kernels to refine gradient computation.
4:     **Optimization.** Update Gaussian kernel parameters (location, density, scale, rotation) using Pixel-Graph-Aware Gradient (PGA).
5:     **if** iteration % 100 == 0 **then**
6:         **Adaptive Control.** Adjust gradient computation strategy based on updated kernel parameters to enhance optimization efficiency.
7:         **Graph Construction.** Construct graph $\mathcal{G}$ using KNN
8:     **end if**
9:     **if** iteration % 500 == 0 **then**
10:       **Density Volumization.** Extract density volumes using a density voxelizer from the radiative representation of the graph structure.
11:       **Dynamic Stopping Criterion.** Evaluate PSNR. Terminate training if PSNR decreases by more than 0.5%.
12:     **end if**
13: **end for**
14: **return** Optimized 3DGS point cloud and density volumes.

---

**Algorithm 3** GR-Gaussian Framework

---

**Input**: Sparse-view X-ray projections
**Output**: Reconstructed 3D volume

1: Perform FDK reconstruction to obtain coarse voxel data.
2: Apply Gaussian filtering for denoising.
3: Construct graph $\mathcal{G}$ using KNN.
4: Optimize Gaussian kernel parameters using Pixel-Graph-Aware Gradient Strategy.
5: Evaluate reconstruction quality using PSNR and SSIM.
6: **return** Reconstructed 3D volume and 3DGS point clouds.

---

# D ADDITIONAL EVALUATION DETAILS

## D.1 DATASET AND METRIC

We evaluate the proposed methods across various modalities, encompassing key CT applications such as medical diagnosis, biological research, and industrial inspection. The dataset consists of 15 cases categorized into three groups: human organs (chest, foot, head, jaw, and pancreas), biological specimens (beetle, bonsai, broccoli, kingsnake, and pepper), and artificial objects (backpack, engine, present, teapot, and mount). The chest and pancreas scans are sourced from LIDC-IDRI and Pancreas-CT, respectively, while broccoli and pepper are obtained from X-Plant Zha et al. (2024). The remaining cases are derived from SciVis. Following the preprocessing protocols in Zha et al. (2024), raw data are normalized to a density range of [0, 1] and resized to volumes of $256 \times 256 \times 256$. Using the tomography toolbox TIGRE, we generate $512 \times 512$ projections spanning a full rotation range of $0°$ to $360°$.

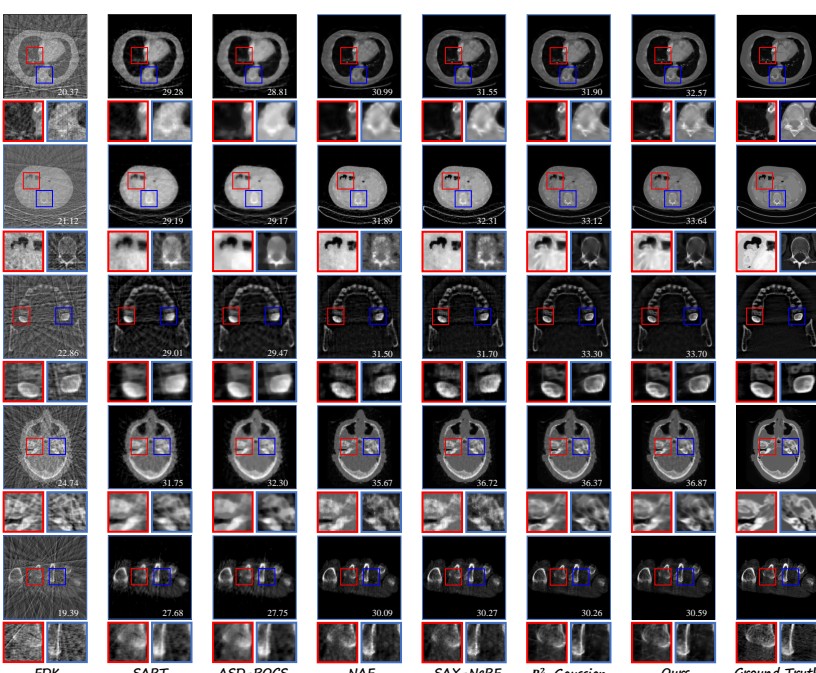

Figure 8: The figure shows **CT reconstructions from the X-3D dataset**, covering human organs (chest, pancreas, jaw, head, and foot) categories. All under 25-view conditions.

We further evaluate our method on real-world data containing scattering effects. We use FIPS Society (2024), a public dataset providing real 2D X-ray projections. FIPS includes three objects (pine, seashell, and walnut). Each case has 721 projections in the range of $0° \sim 360°$. Since ground truth volumes are not available, we use FDK to create pseudo-ground truth with all views and then subsample 25 views for sparse-view experiments.

Reconstruction quality is assessed using PSNR and SSIM, two widely adopted metrics for evaluating image fidelity and structural consistency. PSNR (Peak Signal-to-Noise Ratio) quantifies pixel-wise accuracy by measuring the error between the reconstructed and reference volumes, with higher values indicating better reconstruction quality. SSIM (Structural Similarity Index Measure) evaluates structural coherence across 2D slices in axial, coronal, and sagittal directions, focusing on luminance, contrast, and structural similarity. Together, these metrics provide a comprehensive evaluation of reconstruction performance, balancing pixel-level precision and structural fidelity.

## D.2 EVALUATION

We evaluate GR-Gaussian on the X-3D dataset under 25-view sparse conditions, covering three categories: human organs, biological specimens, and artificial objects. Each category presents distinct reconstruction challenges, ranging from fine anatomical structures to irregular textures and complex geometries.

**Human organs.** Figure 8 shows reconstructions for chest, pancreas, jaw, head, and foot. GR-Gaussian preserves anatomical details and produces accurate density distributions despite the limited number of projections.

**Biological specimens.** Figure 9 presents results for bonsai, beetle, broccoli, kingsnake, and pepper. The framework effectively handles complex textures and irregular shapes, demonstrating robustness in reconstructing non-standard biological structures.

**Artificial objects.** Figure 10 depicts reconstructions of teapot, engine, present, mount, and backpack. GR-Gaussian captures diverse geometries and material properties, highlighting its versatility across different object types.

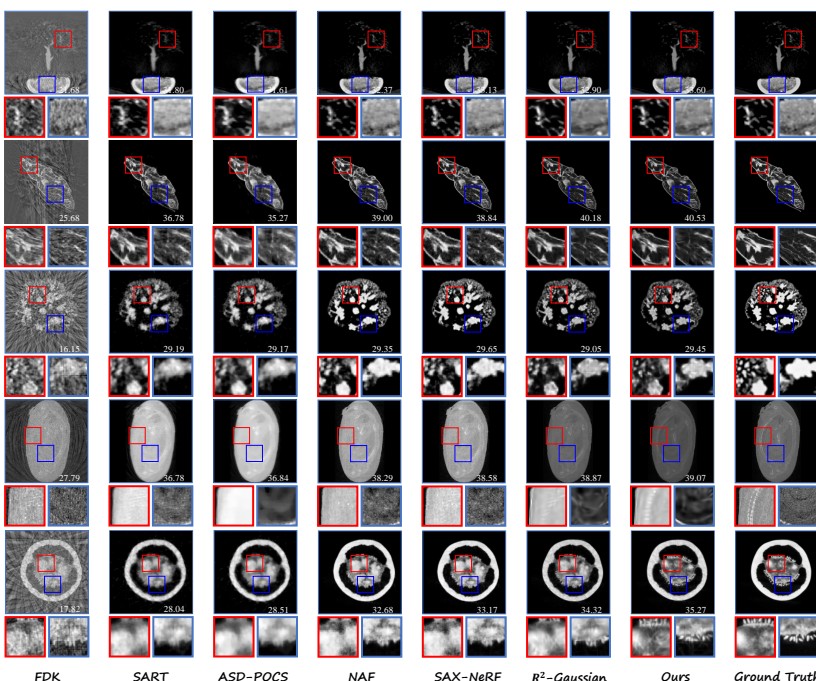

FDK    SART    ASD–POCS    NAF    SAX–NeRF    $R^2$–Gaussian    Ours    Ground Truth

Figure 9: The figure shows **CT reconstructions from the X-3D dataset**, covering biological specimens (bonsai, beetle, broccoli, kingsnake, and pepper) categories. All under 25-view conditions.

**Real-world dataset.** Figure 11 illustrates reconstructions from challenging real-world objects, including seashell, pine, and walnut. Even under 25-view sparse conditions, GR-Gaussian accurately reconstructs fine details, showing robustness against noise and limited projections.

Overall, the results confirm that GR-Gaussian consistently achieves high-fidelity CT reconstructions across diverse categories under sparse-view conditions, validating the effectiveness of graph-based initialization and gradient refinement in practical scenarios.

## E    MORE RESULTS OF ABLATION STUDY

To further validate the effectiveness of our proposed framework, we conducted an ablation study to analyze the impact of individual components and parameter settings on reconstruction quality. Figure visually compares the results obtained by enabling or disabling specific components, demonstrating that the newly introduced modules significantly enhance reconstruction fidelity.

Additionally, we evaluated the influence of key parameter settings, including the SSIM loss weight ($\lambda_{\text{ssim}}$), TV loss weight ($\lambda_{\text{tv}}$), and the number of Gaussians ($M$). The results indicate that our parameter configuration achieves optimal performance, balancing pixel-level accuracy and structural consistency. These findings highlight the importance of both architectural design and parameter tuning in achieving high-quality sparse-view tomographic reconstruction.

Figure provides a comprehensive comparison, illustrating the contributions of each component and parameter to the overall reconstruction quality.

## F    IMPACT OF VIEW COUNT ON RECONSTRUCTION QUALITY

To analyze the effect of view count on reconstruction quality, we conducted experiments using different numbers of sparse-view projections, specifically 10, 15, 20 and 25 views. These experiments aim to evaluate the robustness of the GR-Gaussian framework under varying levels of projection sparsity.

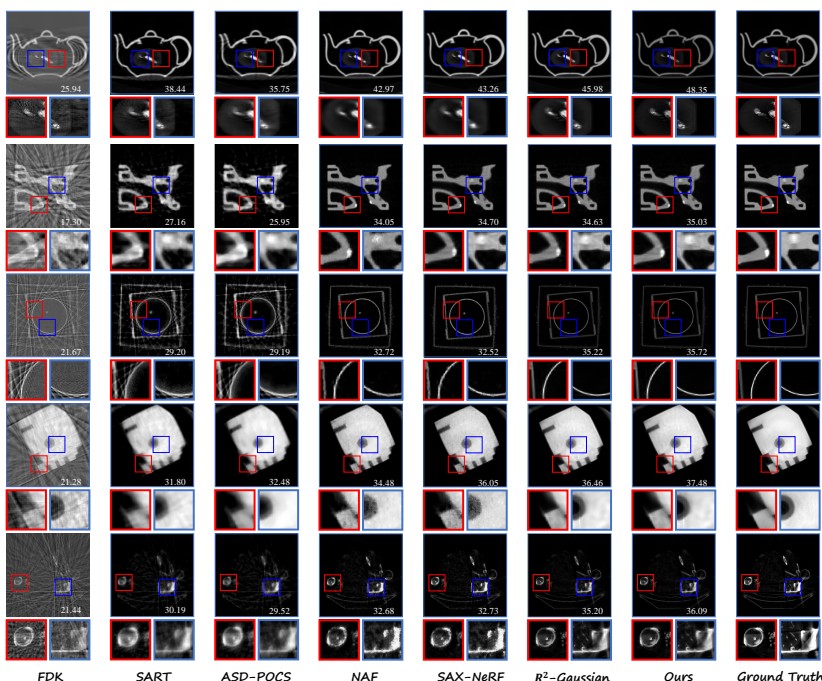

Figure 10: The figure shows **CT reconstructions from the X-3D dataset**, covering artificial objects (teapot, engine, present, mount, and backpack) categories. All under 25-view conditions.

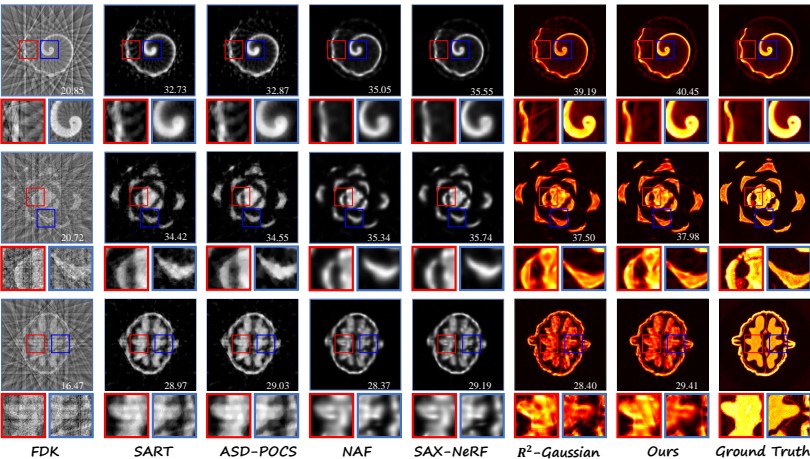

Figure 11: The figure shows **CT reconstructions from the real-world dataset**, covering seashell, pine and walnut) categories. All under 25-view conditions.

## F.1 EXPERIMENTAL SETUP

For each view count, we uniformly sampled projections from the full set of acquired projections spanning a rotation range of $0°$ to $360°$. The tomography toolbox TIGRE was used to process the projections. Reconstruction quality was assessed using PSNR and SSIM metrics, which measure pixel-wise accuracy and structural consistency, respectively.

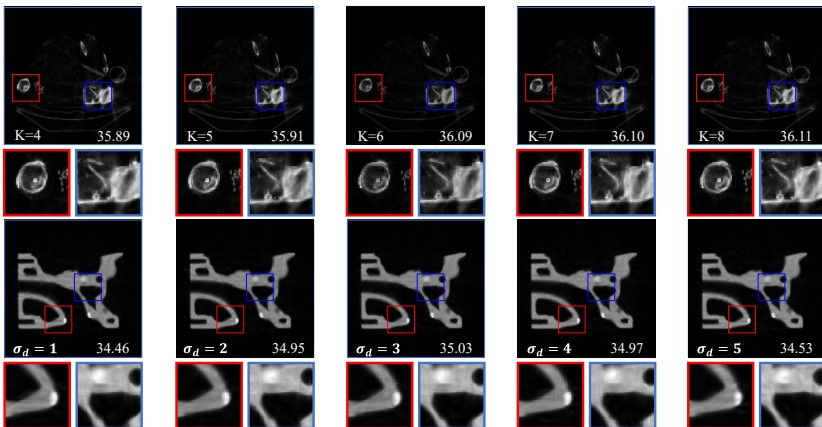

Figure 12: Quantitative comparison of different parameters.

## F.2 RESULTS AND ANALYSIS

Table 6 summarizes the PSNR and SSIM values achieved by GR-Gaussian under different view counts. As expected, reconstruction quality improves with an increasing number of views, as more projection data reduces ambiguity in the reconstructed volume. However, even under extremely sparse conditions (10 views), GR-Gaussian demonstrates competitive performance, achieving reasonable PSNR and SSIM values.

Table 6: Comparison of PSNR and SSIM under different view counts.

| View Count | PSNR (dB) | SSIM |
|---|---|---|
| 10 Views | 28.5 | 0.823 |
| 15 Views | 31.3 | 0.828 |
| 20 Views | 32.1 | 0.834 |
| 25 Views | 35.95 | 0.858 |

The results demonstrate that GR-Gaussian effectively balances reconstruction quality and computational efficiency across varying levels of projection sparsity. While higher view counts naturally lead to better reconstruction fidelity, the framework's ability to achieve competitive performance under extremely sparse conditions (10 views) underscores its robustness and adaptability. This makes GR-Gaussian a practical solution for scenarios where projection data is limited.

## G DISCUSSION OF LIMITATIONS AND FUTURE WORK

## H DISCUSSION OF LIMITATIONS AND FUTURE WORK

While GR-Gaussian demonstrates strong performance across diverse datasets, certain limitations remain. Notably, the current automatic stopping criterion, based on a 5% PSNR adjustment threshold, can fail under real-world CT acquisition conditions. In some cases, reconstructed 3D volumes achieve high PSNR values, yet the corresponding projection PSNR remains low, indicating a mismatch between the density field and actual projection data. This discrepancy is particularly pronounced under high-noise conditions, where noise introduces inconsistencies during rendering, leading to deviations between the reconstructed volume and the measured projections.

Addressing this limitation requires more robust noise-handling mechanisms and a stopping criterion that accounts for projection fidelity in addition to volume PSNR. Future work will focus on integrating advanced denoising techniques and adaptive rendering strategies to better align reconstructed volumes with their corresponding projections. By doing so, we aim to enhance GR-Gaussian's reliability and applicability in real-world sparse-view CT reconstruction scenarios.

# I CONTRIBUTIONS

In this work, we propose the GR-Gaussian framework, which introduces two novel strategies to address the challenges of sparse-view tomographic reconstruction:

- **Denoised Point Cloud Initialization Strategy:** This strategy integrates enhanced FDK reconstruction, Gaussian filtering, and adaptive sampling to generate high-quality initial point clouds. By leveraging graph-based representations, De-Init reduces initialization errors and accelerates convergence, providing a robust foundation for sparse-view reconstruction tasks.

- **Pixel-Graph-Aware Gradient Strategy:** This strategy refines gradient computation by incorporating density differences between neighboring kernels, enhancing optimization accuracy and facilitating kernel splitting when necessary. PGA improves reconstruction fidelity by dynamically adjusting kernel parameters and promoting structural consistency across the reconstructed density field.

Together, these strategies enable GR-Gaussian to achieve accurate and efficient 3D tomographic reconstruction under sparse-view conditions. The framework demonstrates robustness across diverse datasets and experimental scenarios, highlighting its potential for real-world CT applications.

