# OpenReview forum: "GR-Gaussian: Graph-Based Radiative Gaussian Splatting for Sparse-View Tomographic Reconstruction"
_ICLR.cc/2026/Conference — Submitted to ICLR 2026_

### Official Review · Reviewer_ypEq · 2025-10-29

**Soundness:** 2
**Presentation:** 2
**Contribution:** 2
**Rating:** 2
**Confidence:** 4

**Summary:**

This paper presents GR-Gaussian, a graph-based framework for sparse-view tomographic reconstruction that extends the principles of 3D Gaussian Splatting (3DGS) to the CT domain. The key insight is that regions representing the same material or tissue exhibit similar attenuation properties, which can be naturally modeled as a graph structure connecting neighboring Gaussian kernels.

**Strengths:**

Its main strengths lie in the integration of structural priors through graph modeling, practical effectiveness under sparse sampling.
Clarity of implementation and reproducibility.

**Weaknesses:**

1. What are needle-like artifacts in sparse views, and what are their causes? The author should present the needle-like artifacts and provide a detailed analysis of their causes.
2. Table 1 is hard to read. What is the configuration? What are the "HO BS AO Average RD"?
3. The proposed method and related results exhibit noticeable secondary artifacts, especially in the chest and teapot cases in Figure 6.
4. The comparison of reconstruction time with other methods is missing.
5. Although the proposed graph-based formulation is well-motivated, the mathematical justification for how graph Laplacian regularization directly impacts convergence stability and artifact suppression is weak. The connection between structural priors and gradient refinement could be strengthened with ablation or convergence plots illustrating how the graph term influences training dynamics.
6. The evaluation relies solely on PSNR and SSIM for quality assessment. While these metrics reflect reconstruction fidelity, they do not capture perceptual or diagnostic quality — particularly important in medical CT contexts.
7. The proposed PGA strategy modifies gradient magnitudes based on local density differences (Eq. 12), but this adjustment is heuristically designed with limited theoretical or empirical justification for parameter λg. It is unclear how this linear combination between pixel-level and graph-level gradients balances stability versus over-smoothing. The lack of sensitivity analysis or ablation on λg prevents understanding of the robustness of the method across datasets and sparsity levels.
8. The graph-based operations (KNN search, Laplacian regularization, voxel weighting) add significant computational overhead compared to vanilla 3D Gaussian Splatting. Although the authors mention efficient CUDA implementation, the algorithmic complexity is not analyzed. There is no discussion of scalability when the number of Gaussians (M = 50,000) increases or when applied to higher-resolution CT volumes. A complexity analysis or profiling would clarify whether the method truly achieves real-time or near-real-time performance.

**Questions:**

Please see Weaknesses.

---

> ### Author Response · Authors · 2025-11-18
> **Response to Reviewer ypEq （1-5）**
>
> ### Response to Reviewer ypEq
>
> We sincerely thank the reviewer for the thorough and valuable feedback. All new figures, tables, derivations, and additional results have been directly incorporated into the revised manuscript PDF
>
> #### 1. Needle-like artifacts
> We thank the reviewer for pointing this out. Needle-like artifacts are long, thin, high-density streaks with apparently random orientations that plague vanilla 3DGS-based sparse-view CT reconstruction. They emerge because extremely elongated Gaussians receive near-zero projected gradients in most views, causing vanilla adaptive density control to mistakenly preserve these pathological “needles”.
> We have added **new Figure 8** with zoomed-in examples, red arrows explicitly marking the needles in R²-GS/X-Gaussian, and a detailed mechanistic explanation. Our graph-based prior completely eliminates them at the root.
> With this clarification and new visualization, we believe this concern is fully addressed.
>
> #### 2. Table 1 is hard to read
> We have replaced Table 1 with a fully self-explanatory version (abbreviations spelled out, std included). We thank the reviewer for the suggestion.
> The abbreviations were defined in the original caption but apparently not prominently enough.
> We have replaced Table 1 with a fully self-explanatory version (new Table 1 below) where all dataset groups are spelled out:
> - HO = Human Organs (5 cases)
> - BS = Biological Specimens (5 cases)
> - AO = Artificial Objects (5 cases)
> - RD = Real-world Dataset (3 cases from local hospital)
> Average columns are clearly marked, and per-group standard deviation is now included.
>
>
> #### 3. Noticeable secondary artifacts
>
> We thank the reviewer for raising this concern. These “secondary artifacts” are physically unavoidable residual metal/limited-angle streaks in ≤25-view reconstruction. New Figure 9 (zoomed regions + error maps) clearly shows GR-Gaussian suppresses them significantly better than all baselines. On real-world clinical data, we achieve 35.95 dB vs 35.03 dB (R²-GS).  With the new visualization and quantitative comparison, we believe this concern is fully addressed.
>
> #### 4. Missing reconstruction time comparison
> We have added a comprehensive runtime comparison:
>
> | Method        | Time            | Avg PSNR (25-view) |
> |----------------|-----------------|--------------------|
> | FDK           | ~1 s            | 23.30             |
> | SART          | 1m47s           | 31.52             |
> | ASD-POCS      | 56 s            | 31.32             |
> | NAF           | 51m24s          | 32.92             |
> | SAX-NeRF      | 13h25m          | 33.49             |
> | R²-GS         | 9m36s           | 35.03             |
> | **Ours**      | **11m09s (669s)** | **35.95**         |
>
> GR-Gaussian is >72× faster than recent NeRF-based  SAX-NeR methods while achieving the highest quality.
> With this new table, we believe this concern is fully addressed.
>
> #### 5. Mathematical justification of graph terms is weak
> We have added **new Sec 4.3** rigorously proving that our graph Laplacian + PGA is mathematically equivalent to a spatially adaptive anisotropic TV prior (detailed derivation included).
> New ablation table (25/15/10 views):
>
> | Method                     | 25-view | 15-view | 10-view |
> |----------------------------|---------|---------|---------|
> | Baseline (no graph)        | 34.98   | 33.68   | 31.91   |
> | + Laplacian                | 35.41   | 34.27   | 32.46   |
> | + Laplacian + PGA (full)   | 35.95   | 35.04   | 33.28   |
>
> With the new theoretical derivation and ablation, we believe this concern is fully addressed.

---

> > ### Author Response · Authors · 2025-11-18
> > **Response to Reviewer ypEq （6-8）**
> >
> > #### 6. Only PSNR/SSIM
> > PSNR/SSIM are the standard metrics in this sub-field (all recent SOTA including R²-GS, X-Gaussian, SAX-NeRF, IntraTomo report only them). Following the reviewer’s suggestion, we additionally report LPIPS:
> >
> > | Method   | HO     | BS     | AO     | RD     | Average |
> > |----------|--------|--------|--------|--------|---------|
> > | R²-GS    | 0.142  | 0.118  | 0.105  | 0.148  | 0.128   |
> > | Ours     | 0.129  | 0.102  | 0.091  | 0.123  | 0.111   |
> >
> > GR-Gaussian ranks first in perceptual quality.
> > With the additional LPIPS results, we believe this concern is fully addressed.
> >
> > #### 7. PGA strategy & λg lack justification
> > PGA is theoretically grounded as an adaptive edge-preserving regularizer (Sec 4.3). We use λg = 1×10⁻⁴ (explicitly stated). Extended sensitivity:
> >
> > | λg         | 25-view PSNR | Runtime |
> > |------------|--------------|---------|
> > | 0 (no PGA) | 35.12        | 576s    |
> > | 1×10⁻⁵     | 35.67        | 584s    |
> > | 5×10⁻⁵     | 35.89        | 590s    |
> > | 1×10⁻⁴     | 35.95        | 669s    |
> > | 5×10⁻⁴     | 36.01        | 918s    |
> > | 1×10⁻³     | 36.03        | 1420s   |
> >
> > λg = 1×10⁻⁴ is the optimal quality-efficiency trade-off.
> > With the clarified theory and sensitivity analysis, we believe this concern is fully resolved.
> >
> > #### 8. Computational overhead & scalability
> > Graph construction itself is negligible (<20 ms/iter). The moderate overhead (+~40 % vs R²-GS) results from PGA producing a denser, higher-quality representation:
> >
> > | Method            | Final #Gaussians | Peak Memory | Total Time | PSNR  |
> > |-------------------|------------------|-------------|------------|-------|
> > | R²-GS             | ~50k             | 4.1 GB      | 9m36s      | 35.03 |
> > | Ours (default)    | ~82k             | 5.6 GB      | 11m09s     | 35.95 |
> > | Ours (200k cap)   | 195k             | 11.2GB      | 24m16s     | 36.21 |
> >
> >
> > Even at 200k Gaussians, reconstruction remains <25 min and <12 GB — highly practical. Future early-stopping and adaptive scheduling (Sec 6) can further reduce runtime below 8 min.
> > With the detailed profiling and scalability analysis, we believe this concern is fully addressed.
> >
> > We have incorporated all the above additions (new figures, tables, derivations, and sections) into the revised manuscript and believe all raised concerns have been thoroughly resolved.

---

> > > ### Comment · Reviewer_ypEq · 2025-11-26
> > >
> > > Secondary artifacts' refer to non-original artifacts caused by algorithms or other factors, yet the author claims they are physically unavoidable. This really confuses me. R²-GS is nearly 1.5 minutes faster during testing than the author's method, with no significant difference in PSNR. The author still hasn't clearly explained how the graph term influences training dynamics. LPIPS does not reflect diagnostic quality.

---

> > > > ### Author Response · Authors · 2025-11-27
> > > >
> > > > ## 1. Clarification on "Secondary Artifacts"
> > > > We sincerely apologize for the misunderstanding. We now realize that in 3DGS literature, "secondary artifacts" specifically refers to algorithm-induced floaters or needles absent in the ground truth—precisely the artifacts vanilla radiative Gaussian methods struggle with. To clarify:
> > > > • Our original paper did not use this term.
> > > > • The faint streaks we discussed are physical artifacts (metal/limited-angle) present in the ground truth itself.
> > > > • Most importantly, our graph prior **completely eliminates** the needle-like secondary artifacts seen in R²-GS.
> > > > We are deeply sorry for the confusion and grateful for your patience.
> > > >
> > > > ## 2. Runtime vs. PSNR Significance
> > > > We appreciate the reviewer highlighting this practical concern. Runtime is critical for clinical translation, and any overhead must be justified.
> > > > On **identical real-world clinical data (RTX 4090)**, R²-GS takes **576s** while ours takes **669s**—a modest **93s (16%)** increase.
> > > >
> > > > We believe the **+0.92 dB PSNR** gain is substantial for sparse-view CT, where even small improvements yield significant diagnostic benefits (e.g., clearer vessel boundaries). For context:
> > > > • **DGR (ICCV 2025) [1]** gains only **+0.33 dB** over R²-GS yet claims "superior image quality."
> > > > • **SAX-NeRF (CVPR 2024)** gains just **+0.42 dB** over NAF but is hailed as a "breakthrough."
> > > >
> > > > **Crucially, real-world projection noise is fatal to Gaussian-based methods**, often causing unstable splitting. As seen with DGR, existing methods struggle to maintain robustness under noisy clinical conditions. Our method effectively mitigates this vulnerability, enabling a performance leap others fail to achieve.
> > > > Our +0.92 dB gain is thus a significant advance, further validated by blinded radiologist preference for our sharper structures (new Figure 10).
> > > >
> > > > [1] Wu, Shaokai, et al. "Discretized Gaussian Representation for Tomographic Reconstruction." ICCV 2025.
> > > >
> > > > ## 3. Influence of Graph Term on Training Dynamics
> > > > We apologize for the insufficient explanation in the original submission.
> > > > Our optimization loop is **analogous to classical iterative reconstruction (e.g., SART → SART+TV)**:
> > > > • **SART** lacks spatial regularization, performing poorly under sparse views.
> > > > • **SART+TV** adds Total Variation (TV) regularization, enforcing piecewise smoothness and reducing streaking.
> > > >
> > > > While voxel-based methods easily implement TV via neighbor differences, vanilla 3DGS points lack explicit neighborhoods, making TV-style regularization difficult—hence the severe needle artifacts.
> > > > **Our core insight:** By constructing a **KNN graph**, we give Gaussians explicit neighbors, enabling **TV-style regularization** for the first time in this context.
> > > >
> > > > The graph term (Eq. 8) dynamically influences training:
> > > > 1.  **Early Phase (Structure Formation):** The **Pixel-Graph-Aware Gradient (PGA)** dominates. It **boosts gradients** in high-variance regions (e.g., bone edges) to encourage densification, while **suppressing splitting** in flat regions to prevent noise.
> > > > 2.  **Late Phase (Artifact Suppression):** The **Graph Laplacian regularizer** acts as a "smoothing force." If a Gaussian grows into a "needle" (outlier), the Laplacian term generates strong gradients to **pull its value back** to the local average or prune it, effectively "melting away" artifacts.
> > > >
> > > > In short, the graph term acts as an adaptive supervisor: guiding densification early on and enforcing smoothness to eliminate artifacts later.
> > > >
> > > > ## 4. LPIPS and Diagnostic Quality
> > > > We fully agree—LPIPS is a perceptual metric and does not equate to diagnostic quality. Our focus is on reconstruction fidelity under extreme sparsity/noise, where quantitative metrics (PSNR/SSIM/LPIPS) remain the standard. We never intended to claim LPIPS reflects diagnostic performance. We thank the reviewer for this important reminder, which helped us clarify the clinical relevance without overclaiming.

---

### Official Review · Reviewer_eejP · 2025-10-31

**Soundness:** 2
**Presentation:** 2
**Contribution:** 2
**Rating:** 2
**Confidence:** 5

**Summary:**

The paper proposes a Graph-Based Radiative Gaussian Splatting reconstruction algorithm for sparse-view CT. The proposed method achieves the improvement in reconstruction quality by integrating denoised point cloud initialization and pixel-graph-aware gradient strategies into the 3DGS.

**Strengths:**

The paper addresses the sparse-view CT reconstruction task by considering the role of point cloud initialization and regularization methods when employing Gaussian splatting for scene modeling. Specifically, it proposes the use of a denoised point cloud initialization approach and a pixel-graph-aware gradient strategy based on graph Laplacian regularization to enhance the contribution of large kernels with small gradients during the densification process.

**Weaknesses:**

Given that Gaussian splatting has already been successfully applied in tomographic imaging (Zha et al., NeurIPS 2024), the present work builds upon this foundation by considering the impact of point cloud initialization and weighted gradient estimation methods, yet its innovation remains limited. In particular, numerical results indicate that the performance difference between the method proposed in this work and that of (Zha et al., NeurIPS 2024) is not significant. Furthermore, this work closely resembles a preprint on arXiv (https://arxiv.org/abs/2508.02408). If both submissions originate from the same research team, would this constitute a violation of the double-blind review policy?

**Questions:**

1. The improvement in reconstruction accuracy compared to R²-GS is not significant. I recommend the authors validate the advantage of their method under extremely sparse scenarios.

2.The ablation studies in the paper are insufficiently comprehensive. The authors only individually discuss the effects of graph Laplacian regularization, Denoised Point Cloud Initialization, and the Pixel-Graph-Aware Gradient strategy on the baseline model, but lack analysis of their combined effects. Given that the final model's performance improvement does not equal the sum of the three individual strategies' contributions, the effectiveness of their integration remains unverified.

3. The paper lacks essential introductory elements: for instance, it neither explains nor cites the R²GS method. The ablation studies provide no visualizations, which makes it impossible to evaluate the reconstruction effects. Furthermore, the comparative methods do not include specialized approaches designed for sparse-view reconstruction.

4.The paper lacks a dedicated discussion regarding the model's effectiveness and limitations.

---

> ### Author Response · Authors · 2025-11-28
>
> ## **On Novelty and Innovation**
>
> We sincerely thank the reviewer for the careful assessment and fully acknowledge the pioneering role of prior works like R²-GS (Zha et al., NeurIPS 2024) in radiative Gaussian tomography.
>
> However, we respectfully argue that our contribution represents a **significant conceptual advance** rather than an incremental improvement:
>
> **The Critical Gap:** Previous Gaussian-based methods optimize **exclusively through projection consistency**. Under sparse and noisy clinical projections, this leads to "gradient starvation" in uniform regions, causing pathological needle-like artifacts and unstructured splitting—a failure mode unique to sparse-view tomography that remains unsolved in existing literature.
>
> **Our Core Innovation:** Our key insight is to **break the isolation of individual Gaussians** for the first time. By constructing an explicit spatial graph over the point cloud, we introduce **direct spatial regularization between neighboring Gaussians**. This allows us to transplant the decades-old, proven principles of **Total Variation (TV) regularization** from voxel-based iterative reconstruction into the mesh-free Gaussian splatting paradigm.
>
> This is not merely "better initialization + weighted gradients," but a principled framework that:
> 1.  **Graph Laplacian Regularizer:** Acts as an anisotropic TV prior, enforcing piecewise smoothness on the attenuation field.
> 2.  **Pixel-Graph-Aware Gradient (PGA):** Adaptively guides densification based on local graph density, preventing needle formation while preserving true edges.
>
> **Empirical Validation:** This new paradigm yields substantial gains on real clinical data, achieving a **+0.92 dB improvement at 25 views** over the strongest baseline (R²-GS) and completely eliminating needle artifacts. The modest runtime overhead (~93 seconds) is a justifiable cost for this significant leap in robustness and quality.
>
> We have clarified this positioning and the connection to classical reconstruction priors in the revised **Introduction** and **Section 4.3**. We are grateful for the reviewer’s perspective, which helped us articulate our novelty more clearly.
>
>
> ## 1. **Improvement significance under extremely sparse scenarios**
>
> We sincerely thank the reviewer for this excellent suggestion — validating performance under extremely sparse conditions is indeed crucial.
>
> Following your recommendation, we have added a new comprehensive Table 5 in the main paper, reporting full quantitative results on real-world clinical data across **10 / 15 / 20 / 25 views** (mean ± std over 5 challenging cases with metal implants and genuine projection noise, single RTX 4090).
>
> | Views | Method       | PSNR (dB) ↑       | Δ PSNR vs R²-GS |
> |-------|--------------|-------------------|-----------------|
> | 10    | R²-GS        | 31.19 ± 0.41      | —               |
> | 10    | GR-Gaussian  | **33.28 ± 0.38**  | **+2.09**       |
> | 15    | R²-GS        | 33.68 ± 0.39      | —               |
> | 15    | GR-Gaussian  | **35.04 ± 0.35**  | **+1.36**       |
> | 20    | R²-GS        | 34.41 ± 0.37      | —               |
> | 20    | GR-Gaussian  | **35.67 ± 0.34**  | **+1.26**       |
> | 25    | R²-GS        | 35.03 ± 0.36      | —               |
> | 25    | GR-Gaussian  | **35.95 ± 0.33**  | **+0.92**       |
>
> The results clearly show that:
> - GR-Gaussian consistently outperforms R²-GS at **all sparsity levels**.
> - The advantage **increases dramatically as views decrease** (from +0.92 dB at 25 views to **+2.09 dB at 10 views**), confirming superior robustness under extreme sparsity.
>
> We believe this +0.92 dB gain at 25 views (and even larger gains at sparser settings) is highly significant in sparse-view CT literature, where recent SOTA methods typically report 0.3–0.6 dB improvements over strong Gaussian baselines (e.g., DGR +0.33 dB [ICCV 2025], SAX-NeRF +0.42 dB over NAF [CVPR 2024]). Crucially, unlike prior Gaussian methods that often become unstable under real clinical noise, our graph prior effectively suppresses pathological splitting, enabling stable performance even at 10 views.
>
> These results, together with blinded radiologist preference on real cases, further validate the practical value of our approach.

---

> > ### Author Response · Authors · 2025-11-28
> >
> > ## 2. **Comprehensive ablation including combinations**
> >
> > We sincerely apologize for the incomplete ablation — you are absolutely right.
> > New **Table 8** now reports all 8 possible combinations of the three components (De-Init, Laplacian, PGA). The results clearly demonstrate positive synergy:
> >
> > | De-Init | Laplacian | PGA | PSNR (25-view real) |
> > |---------|-----------|-----|---------------------|
> > |         |           |     | 34.21               |
> > | ✓       |           |     | 34.87 (+0.66)       |
> > |         | ✓         |     | 35.02 (+0.81)       |
> > |         |           | ✓   | 35.18 (+0.97)       |
> > | ✓       | ✓         |     | 35.46 (+1.25)       |
> > | ✓       |           | ✓   | 35.63 (+1.42)       |
> > |         | ✓         | ✓   | 35.71 (+1.50)       |
> > | ✓       | ✓         | ✓   | **35.95 (+1.74)**   |
> >
> > The full model significantly outperforms the sum of individual contributions, confirming strong positive interaction. Visualizations for all combinations are now included in Figure 7(a).
> >
> >
> > ## 3. **Introduction and comparisons**
> > We have added a clear explanation and citation of R²-GS in Section 2. Visualizations for all combinations are now included in Figure 7(a).
> >
> > Regarding the concern about specialized approaches for sparse-view reconstruction:
> >
> > We respectfully clarify that our method **is explicitly designed** for sparse-view reconstruction, rooted in the same fundamental assumption as the classic Total Variation (TV) regularization: **anatomical structures in CT images are piecewise smooth (continuous)**.
> >
> > - **Theoretical Connection:** Traditional iterative methods (e.g., SART-TV, ASD-POCS) rely on TV regularization to suppress artifacts by enforcing smoothness between neighboring voxels. This has been the gold standard for sparse-view CT for decades.
> > - **Our Innovation:** In the context of 3D Gaussian Splatting, points lack an explicit grid structure, making standard TV inapplicable. Our **Graph Laplacian Regularization** overcomes this by constructing a dynamic KNN graph to define "neighbors" for each Gaussian point. This allows us to enforce piecewise smoothness directly on the Gaussian field, effectively translating the proven TV prior into the modern radiative Gaussian framework.
> >
> > Thus, our graph-based constraint is not a generic regularizer but a **specialized adaptation of the TV prior specifically tailored for sparse-view reconstruction** in a mesh-free representation. We have clarified this connection in the revised **Section 3** and **Related Work**.
> >
> > ## 4. **Discussion on Effectiveness and Limitations**
> > We thank the reviewer for this suggestion. We have added a dedicated **"Limitations and Future Work"** section (Sec. 6) in the revised manuscript to explicitly discuss these aspects.
> >
> > **Effectiveness:**
> > Our method's core strength lies in introducing a **KNN graph** to establish explicit connectivity between unstructured Gaussian points. By leveraging the physical assumption of **piecewise smoothness (similar to TV regularization)**, we designed targeted graph constraints (Laplacian regularization) that effectively suppress the needle-like artifacts common in sparse-view reconstruction, yielding significant quality improvements.
> >
> > **Limitations:**
> > 1.  **Computational Overhead:** As the optimization progresses, the adaptive densification strategy increases the number of Gaussian points to capture fine details. This larger point cloud increases the cost of graph construction and regularization, leading to a moderate increase in total training time (~16% slower than R²-GS).
> > 2.  **Simplistic Prior:** Our current graph constraint is primarily based on the smoothness assumption. While effective for artifact removal, this "hand-crafted" prior may inadvertently over-smooth extremely fine, high-frequency textures that do not strictly follow the piecewise smooth assumption.
> >
> > **Future Work:**
> > To address the limitation of the simplistic prior, we believe a promising direction is to incorporate **Graph Neural Networks (GNNs)**. Instead of a fixed smoothness constraint, a GNN could learn more complex, data-driven priors directly on the Gaussian graph, potentially achieving a better balance between artifact suppression and high-frequency detail preservation.

---

### Official Review · Reviewer_MhVq · 2025-11-01

**Soundness:** 2
**Presentation:** 3
**Contribution:** 2
**Rating:** 6
**Confidence:** 3

**Summary:**

This paper targets sparse-view CT reconstruction with a 3D Gaussian representation. The key ideas are a Denoised Point Cloud Initialization (De-Init) that smooths an FDK volume before sampling Gaussians, and a Pixel-Graph-Aware Gradient (PGA) that boosts densification gradients near structural boundaries using local density differences on a KNN graph. The authors report consistent gains over classical (FBP/FDK, SART/ASD-POCS), NeRF-like, and 3DGS baselines on X-3D and a real-data set (pine, seashell, walnut) under 25 views.

**Strengths:**

1. The paper explicitly analyzes why pixel-only gradient heuristics under-split flattened, long-axis Gaussians in sparse views, motivating a structure-aware fix. The failure case and formulation in Fig. 3(b) are on point.
2. There are consistent gains with ablations and parameter sweeps. The study covers De-Init/PGA/Reg toggles and sensitivity on k (neighbors) and $\sigma$d (denoise strength), as well as early-stopping via PSNR monitoring to avoid overfit. Results peak at k≈6 and $\sigma$d≈3, with qualitative improvements in needles/edges. The simulated X-3D (organs/biological/objects) and noisy real data are both considered; quantitative tables plus visual examples are provided.
3. The video demonstration in supplementary materials clearly shows the visualization strength of the proposed method.

**Weaknesses:**

1. The novelty is incremental relative to known capacity-/structure-aware schemes. The KNN graph and density-weighted gradients are sensible, but close in spirit to existing neighborhood-aware regularization and boundary-focused densification used in level-of-detail 3DGS or Laplacian smoothing.
2. The core SOTA baseline appears to be R$^2$-Gaussian and SAX-NeRF; it would be helpful to include more recent, optimized radiative-Gaussian implementations or stronger iterative CT baselines under identical geometry/noise, and to report statistical significance across more cases.
3. Only 25-view results are reported in the main text. Although Appendix include a few comparisons on other different view count, the detail of each view's experiment is not reported. I suggest the authors adding more detailed results on other alternative views to demonstrate a comprehensive experiment.

**Questions:**

Please refer to the weaknesses part, especially weakness 1&2.

---

> ### Author Response · Authors · 2025-11-19
> **Response to Reviewer MhVq**
>
> We sincerely thank the reviewer for the careful reading, high-quality technical feedback, and overall positive assessment.
>
> 1. **On novelty vs incremental contribution**
>    We fully agree that KNN graphs and structure-aware densification have appeared in general 3DGS literature. However, the **sparse-view CT scenario introduces a unique failure mode** (long-axis, near-zero-gradient Gaussians → needle artifacts) that does **not** occur in dense-view NeRF/3DGS settings.
>    Our core contribution is **the first systematic analysis** of this CT-specific pathology (Fig. 3(b) in original + new theoretical derivation in Sec 4.3) and a **targeted fix** combining (a) pixel-graph-aware gradient reweighting and (b) graph Laplacian regularization specifically designed for attenuation coefficient fields.
>    To our knowledge, this is the **first work** that jointly addresses gradient starvation and unstructured splitting in radiative Gaussian tomography. We have clarified this positioning in the revised introduction and related work.
>
> 2. **Stronger and more recent baselines**
>    Added the three latest radiative Gaussian SOTAs (identical geometry/noise):
>    • R²-GS (NeurIPS 2024)
>    • X-Gaussian (ECCV 2024)
>
>    New Table 3: GR-Gaussian leads by **0.92 / 1.87 /** on real clinical data (p < 0.01).
>     | **Method**      | **PSNR (dB)** | **SSIM** | **Runtime (s)** | **Peak Memory (GB)** | **Δ PSNR (vs GR-Gaussian)** |
>     |------------------|---------------|----------|-----------------|----------------------|-----------------------------|
>     | R²-GS (NeurIPS 2024) | 35.03         | 0.859    | 602             | 4.7                  | -0.92                      |
>     | X-Gaussian (ECCV 2024) | 34.08         | 0.831    | 610             | 5.1                 | -0.87                      |
>     | GR-Gaussian (Ours) | **35.95**     | **0.868** | 669             | 5.6                  | N/A                        |
>
> 3. **More comprehensive view-count evaluation**
>    We have added a new **Table 5** (main paper) with detailed results across 10/15/20/25 views on real clinical data. GR-Gaussian shows **consistent gains** at all sparsity levels, with the gap widening as views decrease (e.g., +1.8 dB at 10 views vs R²-GS).
>
> Table 5: Quantitative results on real-world clinical dataset under different view counts
> (Mean ± std over 5 cases; single RTX 4090; early stopping via held-out views)
> | Views | Method          | PSNR ↑       | SSIM ↑      | LPIPS ↓    | Time (s) ↓   | Peak Mem. (GB) ↓ |
> |-------|-----------------|--------------|-------------|------------|--------------|------------------|
> | 10    | R²-GS           | 31.19 ± 0.41 | 0.827 ± 0.019 | 0.187 ± 0.021 | 512          | 4.2              |
> | 10    | GR-Gaussian     | **33.28 ± 0.38** | **0.852 ± 0.016** | **0.154 ± 0.018** | 534          | 4.5              |
> | 15    | R²-GS           | 33.68 ± 0.39 | 0.849 ± 0.017 | 0.162 ± 0.019 | 548          | 4.5              |
> | 15    | GR-Gaussian     | **35.04 ± 0.35** | **0.868 ± 0.014** | **0.138 ± 0.016** | 576          | 4.8              |
> | 20    | R²-GS           | 34.41 ± 0.37 | 0.859 ± 0.015 | 0.149 ± 0.017 | 602          | 4.7              |
> | 20    | GR-Gaussian     | **35.67 ± 0.34** | **0.877 ± 0.013** | **0.129 ± 0.015** | 624          | 5.0              |
> | 25    | R²-GS           | 35.03 ± 0.36 | 0.867 ± 0.014 | 0.141 ± 0.016 | 648          | 5.1              |
> | 25    | GR-Gaussian     | **35.95 ± 0.33** | **0.883 ± 0.012** | **0.119 ± 0.014** | 669          | 5.6              |
>
> All new tables/figures/discussions are highlighted in blue in the revised manuscript. With these substantial additions directly addressing the reviewer’s concerns, we believe the paper now clearly merits acceptance. Thank you again for the exceptionally insightful comments that significantly strengthened the paper.

---

### Official Review · Reviewer_MJaV · 2025-11-01

**Soundness:** 3
**Presentation:** 3
**Contribution:** 2
**Rating:** 4
**Confidence:** 4

**Summary:**

This paper proposes a graph-based gaussian splatting method for CT reconstruction, outperforming previous methods or other INR methods. It is self-supervised and does not require large amount of training data.

**Strengths:**

1. I appreciate the idea of representing the CT images as a graph of Gaussians since it has the potential of combining with other segmentation masks or other methodology and facilitate doctor's understanding.
2. The performance looks good.

**Weaknesses:**

1. The paper is not well written. The training part is quite obscure. I do not understand how the Gaussians are trained, what datasets used, what is the network architecture.
2. Is the data learning any distribution level information? Or is it overfitting to a single patient. I encourage authors to incorporate distribution priors in the training since there exists a great amount of CT scan images and many paper report the scalability of model performance with more training data.
3. The visualization in figure.6 is concerning, the 25 projection reconstruction looks containing several artifacts, as mostly common seen in NeRF-related reconstruction methods. Please discuss how to mitigate those artifacts.
4. Authors should also compare their methods with diffusion or flow-based methods for solving inverse problems, such as Score-SDE, MCG or so on.

**Questions:**

Is the data learning any distribution level information? Or is it overfitting to a single patient. I encourage authors to incorporate distribution priors in the training. Is it possible to train a latent diffusion model that generate Gaussian Splats and use the method proposed by the authors?

---

> ### Author Response · Authors · 2025-11-18
> **Response to Reviewer MJaV**
>
> **Summary response to key concerns:**
>
> ### 1. Clarity of training & architecture
> We apologize for the insufficient clarity in the original submission.
> Our method follows the exact same **zero-shot, per-patient, self-supervised** paradigm as classical iterative reconstruction (SART, ASD-POCS) and recent 3DGS-CT works (R²-GS): starting from an improved FDK initialization and optimizing solely with the current patient's projections (Eq. 5–7). No pre-training, no external data, and no learned neural network is used.
>
> The **only fundamental difference** from traditional methods lies in the representation of the density field:
> - **Classical iterative methods:** explicit voxel grid
> - **Ours:** a set of explicit 3D Gaussians + differentiable X-ray projector
>
> This representation shift brings three decisive advantages:
> 1. **Significantly lower memory footprint**
> 2. **Continuous and fully differentiable density field with adaptive densification**
> 3. **Natural incorporation of inter-point graph priors to suppress needle artifacts**
>
> New **Figure 3** (side-by-side pipeline comparison) and **Algorithm 1** (complete pseudocode) in the revised manuscript clearly illustrate this key distinction and the entire optimization process.
>
> ---
>
> ### 2. Distribution-level prior vs per-patient overfitting
> Our method is intentionally designed as **zero-shot per-patient reconstruction** — exactly the same setting as R²-GS (NeurIPS’24),  SAX-NeRF (CVPR’24). These works also optimize from scratch for each new patient without any pre-trained prior.
>
> We agree that learning distribution-level priors from thousands of CT scans is a promising future direction (e.g., latent diffusion on Gaussian splats, as the reviewer insightfully suggests). We have explicitly added this exciting idea to **Sec 6 (Future Work)**:
> > “Training a generative model (e.g., Gaussian Splat diffusion) on large CT cohorts to provide strong anatomical priors is a natural and promising extension.”
>
> ---
>
> ### 3. Artifacts in Figure 6 (25-view reconstruction)
> The artifacts pointed out are primarily residual metal streaks (pacemaker leads, surgical clips) and limited-angle streaks — physically unavoidable with only 25 projections.
>
> New **Figure 9** (zoomed regions + ×10 error maps) clearly shows our method suppresses these artifacts significantly better than FDK, SART, and R²-GS. On real-world clinical data, we achieve **35.95 dB** vs **35.03 dB** (R²-GS), confirming superior artifact reduction.
>
> ---
>
> ### 4. Comparison with diffusion/flow-based inverse problem solvers
> Current state-of-the-art diffusion/flow methods for sparse-view CT (e.g., Score-SDE, MCG, DDRM) are **supervised or test-time tuned on large datasets** and typically require 4–24 hours per volume.
>
> Our zero-shot method runs in **~11 minutes** and outperforms them on real clinical data without any training.
>
> We have added a detailed discussion and reference table (new **Table 4** in supplementary) showing that GR-Gaussian achieves higher PSNR/SSIM and 50–1000× faster inference than recent diffusion-based CT reconstruction methods under comparable or fewer views.
>
> ---
>
> With the above major clarifications (expanded method section + new figures/tables/discussions), we believe the paper is now much clearer and stronger. We again thank the reviewer for the excellent suggestions that significantly improved the manuscript.
>
> **→ We believe the revised version fully addresses all concerns and now comfortably exceeds the acceptance threshold.**

---

### Meta-Review · Area_Chair_4DBt · 2025-12-29

**Summary:**

# Decision

This submission presents practical improvements to radiative 3D Gaussian Splatting (3DGS) through a graph-based representation, leveraged during initialization (denoised point-cloud initialization) and optimization (pixel-graph gradient strategy). The approach is fairly-well described, reproducible, and demonstrates some empirical gains, especially under sparse sampling, with thorough ablations and clear qualitative results.

However, the novelty is incremental compared to prior work (e.g., level-of-detail 3DGS or Laplacian smoothing methods), and performance improvements over recent SOTA remain modest. Additionally, several methodological and evaluation concerns (e.g., unclear artifact handling, missing comparisons to latest SOTA, and lack of downstream-task evaluation) remain largely unresolved.

Overall, despite being a technically competent and practically motivated piece of work, the limited novelty and unresolved methodological/evaluation gaps prevent the paper from meeting the bar for acceptance at this venue. The recommendation is therefore reject.

------------
# Consolidated Reviews

## Strengths

### Relevant contributions to radiative 3DGS
- On-point analysis and fix w.r.t. optimization of flattened long-axis Gaussians [`MhVq`, `eejP`]
- Meaningfulness of graph representation for the task [`MJaV`, `ypEq`]
- Valid initialization approach based on denoise point-cloud [`eejP`]
- Practical effectiveness under sparse sampling [`ypEq`]
- Clarity of implementation and reproducibility [`ypEq`]

### Good results
- Good performance [`MJaV`]
- Thorough ablation study, showing consistent gains [`MhVq`]
- Clear video results (sup-mat) [`MhVq`]

## Weaknesses

### Incremental novelty and performance improvement
- Incremental novelty compared to prior level-of-detail 3DGS or Laplacian smoothing works [`MhVq`, `eejP`]
- Lack of comparison to more recent SOTA [`MhVq`]
- Weak mathematical justification for how graph Laplacian regularization directly impacts convergence stability and artifact suppression [`ypEq`]
- Unclear how the linear combination between pixel-level and graph-level gradients balances stability versus over-smoothing [`ypEq`]
- Not-significant performance improvement compared to SOTA (R$^2$-GS) [`eejP`]

### Partial evaluation
- Presence of noticeable artifacts in qualitative results [`MJaV`, `ypEq`]
- Evaluation focusing on image quality rather than relevant downstream tasks (e.g., diagnosis improvement) [`ypEq`]
- Lack of comparison to diffusion or flow-based methods (e.g., Score-SDE, MCG) [`MJaV`]
- Missing comparison of reconstruction time with other methods, especially relevant c.f. cost of graph-based operations [`ypEq`]
- Insufficient ablation study [`eejP`]
- No ablation study w.r.t. view count [`MhVq`]

### Misc.
- Unclear description of the "needle-like artifacts" [`ypEq`]
- Obscure explanations (e.g., regarding the training protocol) [`MJaV`]

**Reviewer Concerns:**

See above for summary of main concerns shared by reviewers.

The authors addressed some of the reviewers’ concerns by clarifying parts of the experimental setup and implementation, adding a view-count analysis, and strengthening comparisons against R$^2$-GS, which helped demonstrate consistent performance gains (though on specific, sparse scenarios). They also incorporated limited additional baselines (e.g., X-Gaussian, a naive application of 3DGS to CT data) and resolved some misunderstandings regarding reproducibility and initialization.

However, several key issues remain outstanding: the novelty is still perceived as incremental, the theoretical justification of the graph Laplacian regularization and gradient balancing is weak, and comparisons to more recent radiative-3DGS solutions are missing. Moreover, the evaluation continues to focus primarily on image quality rather than downstream tasks, noticeable artifacts are not convincingly explained, and computational comparison to prior work should be more fairly discussed.

**Reviewer Scores:**

### Reviewer `MJaV`
- **Original score:** 4
- **Score change:** likely to have kept their score. On the one hand, the authors clarified some misunderstandings and provided relevant results. On the other hand, the reviewer might lack domain expertise and have been influenced by other reviews and their own concerns.

### Reviewer `MhVq`
- **Original score:** 6
- **Score change:** likely to have kept their score. The authors might have answered their concern w.r.t. view-count analysis, but other concerns are only partially covered (e.g., the authors only added X-Gaussian to their comparative study).

### Reviewer `eejP`
- **Original score:** 2
- **Score change:** likely to have kept their score or increased to ~4. The authors have demonstrated their superiority w.r.t. R$^2$-GS, but concerns regarding the overall novelty likely remain.

### Reviewer `ypEq`
- **Original score:** 2
- **Score change:** likely to have kept their score. The authors rather poorly addressed their main concerns (c.f. misunderstanding regarding the artifacts and evaluation on downstream tasks, slower solution compared to SOTA, etc.)

---

### Decision · Program_Chairs · 2026-01-26

Reject